# Structural basis for activation and non-canonical catalysis of the Rap GTPase activating protein domain of plexin

Yuxiao Wang[1†], Heath G Pascoe[1†], Chad A Brautigam[2], Huawei He[1‡], Xuewu Zhang[1,2*]

[1]Department of Pharmacology, University of Texas Southwestern Medical Center, Dallas, United States; [2]Department of Biophysics, University of Texas Southwestern Medical Center, Dallas, United States

**Abstract** Plexins are cell surface receptors that bind semaphorins and transduce signals for regulating neuronal axon guidance and other processes. Plexin signaling depends on their cytoplasmic GTPase activating protein (GAP) domain, which specifically inactivates the Ras homolog Rap through an ill-defined non-canonical catalytic mechanism. The plexin GAP is activated by semaphorin-induced dimerization, the structural basis for which remained unknown. Here we present the crystal structures of the active dimer of zebrafish PlexinC1 cytoplasmic region in the apo state and in complex with Rap. The structures show that the dimerization induces a large-scale conformational change in plexin, which opens the GAP active site to allow Rap binding. Plexin stabilizes the switch II region of Rap in an unprecedented conformation, bringing Gln63 in Rap into the active site for catalyzing GTP hydrolysis. The structures also explain the unique Rap-specificity of plexins. Mutational analyses support that these mechanisms underlie plexin activation and signaling.

*For correspondence: xuewu.zhang@utsouthwestern.edu

†These authors contributed equally to this work

‡Present address: State Key Laboratory of Silkworm Genome Biology, Southwest University, Chongqing, China

Competing interests: The authors declare that no competing interests exist.

## Introduction

Plexins are a large group of type I transmembrane proteins that serve as the major receptors for semaphorins (*Yazdani and Terman, 2006*; *Tran et al., 2007*). Plexin-mediated semaphorin signaling controls neuronal axon guidance as well other essential processes such as angiogenesis and immune responses (*Sakurai et al., 2012*; *Takamatsu and Kumanogoh, 2012*). Aberrant plexin/semaphorin signaling has been implicated in numerous pathologies including neurological disorders and cancer (*Yaron and Zheng, 2007*; *Tamagnone, 2012*; *Gu and Giraudo, 2013*). Plexins all possess a large multi-domain extracellular region, a single transmembrane helix and a multi-domain cytoplasmic region. Binding of semaphorin to the extracellular region of plexin triggers activation of the cytoplasmic region, which relays the signal to downstream pathways.

The plexin cytoplasmic region contains a juxtamembrane segment, a RhoGTPase binding domain (RBD) and a GTPase activating protein (GAP) domain (*Rohm et al., 2000*; *Hu et al., 2001*; *He et al., 2009*; *Tong et al., 2009*; *Bell et al., 2011*). The juxtamembrane segment has been suggested to regulate plexin signaling by interacting with the GAP domain or mediating oligomerization (*He et al., 2009*; *Bell et al., 2011*). Binding of RhoGTPases such as Rac1 and RND1 to the RBD facilitates plexin activation (*Vikis et al., 2000*; *Driessens et al., 2001*; *Zanata et al., 2002*; *Turner et al., 2004*; *Tong et al., 2007*), the mechanism of which is not well understood (*Bell et al., 2011*; *Wang et al., 2012*). The GAP domain in plexin shows structural homology to RasGAPs such as p120GAP, and contains a functionally essential arginine residue corresponding to the catalytic 'arginine finger' in RasGAPs (*Rohm et al., 2000*; *Oinuma et al., 2004*; *He et al., 2009*). Plexins have been reported previously to be GAPs for the Ras homologs R-Ras and M-Ras (*Oinuma et al., 2004*; *Saito et al., 2009*). Our recent study, however, has demonstrated that the plexin GAP does not act directly on R-Ras or M-Ras (*Wang et al., 2012*). Instead

**eLife digest** A key question in neurobiology is how the brain becomes wired up. How do axons—the 'wires' along which neural signals flow—know in which direction to grow to reach their intended targets? A family of signalling proteins called semaphorins contribute to this process by acting as stop signals for axons that are heading in the wrong direction. The actions of semaphorins are mediated by receptors known as plexins, which are found on the membranes of axons.

Plexins contain an extracellular domain that binds semaphorin, and a large domain inside the cell that can turn semaphorin binding into cellular responses. When a semaphorin protein binds to the extracellular domain of a plexin receptor, the domain inside the cell joins with the intracellular domain of a neighbouring receptor to form a dimer. This activates the intracellular domain, which turns on its ability to inactivate a molecule called Rap. The end result is that the axon stops growing and changes direction, but the molecular mechanisms through which these events occur are not well understood.

Now, Wang, Pascoe et al. have worked out the structure of the dimers formed by the intracellular plexin domains, both alone and in complex with Rap. The structures reveal how the dimer drives a shape change of the intracellular domain to enable it to bind Rap, and show that Rap itself adopts a novel conformation upon binding to plexin. This conformational change in Rap catalyses the breakdown of a signalling molecule called GTP, which inactivates Rap and triggers an intracellular signalling cascade that causes the axon to collapse and change direction.

Lastly, Wang, Pascoe et al. have shown that the highly specific nature of these interactions depends on particular amino-acid residues in both Rap and the plexin receptor. Further work is now required to determine whether this pattern of activation represents a general mechanism for signalling by plexin receptors, and for the inhibition of Rap.

it is active specifically to the Ras homolog Rap, and this RapGAP activity is critical for plexin signaling. GTP-bound active Rap is a key activator of integrin for promoting cell–matrix adhesion (*Gloerich and Bos, 2011*). Conversion of Rap into the GDP-bound inactive form by the plexin GAP likely contributes to plexin-mediated repulsive axon guidance and other cell morphological changes through causing inactivation of integrin and weakening cell–matrix adhesion (*Wang et al., 2012*).

RasGAPs such as p120GAP and neurofibromin facilitate GTP hydrolysis of Ras, R-Ras and M-Ras by providing the conserved arginine finger to stabilize the leaving γ-phosphate group (*Li et al., 1997*; *Scheffzek et al., 1997*; *Scheffzek et al., 1998*; *Quilliam et al., 1999*; *Ohba et al., 2000*; *Bos et al., 2007*). Concomitantly, a conserved glutamine in the GTPases (Gln61 in Ras) coordinates the nucleophilic water for hydrolysis. Many other GAP/small GTPase pairs use similar mechanisms to catalyze GTP hydrolysis (*Bos et al., 2007*). Rap is distinct from Ras/R-Ras/M-Ras in that it has a threonine at position 61, which lacks the ability to coordinate the catalytic water. Canonical RapGAPs are structurally unrelated to RasGAPs and catalyze Rap GTP hydrolysis by providing an asparagine residue (referred to as the 'Asn thumb') to fulfill the water coordination role of Gln61 in Ras (*Scrima et al., 2008*). SynGAP (Synaptic GAP), and three GAP1 family members Rasal (Ras-GTPase-activating-like protein), CAPRI (Ca$^{2+}$-promoted Ras inactivator) and GAP1$^{IP4BP}$ (tetrakisphosphate binding protein) are dual-specificity GAPs, active to both Ras and Rap. Plexins and these dual-specificity GAPs share the RasGAP fold that contains the arginine finger but lack a conserved Asn thumb (*Kupzig et al., 2006*; *Pena et al., 2008*). They facilitate GTP hydrolysis for Rap through a distinct, poorly understood mechanism. A recent study has suggested that Gln63 in Rap plays a role analogous to Gln61 in Ras in the non-canonical catalysis of the dual-specificity GAPs (*Sot et al., 2010*). Mutating Gln63 in Rap abolishes GTP hydrolysis catalyzed by both the dual-specificity GAPs and plexins (*Sot et al., 2010*; *Wang et al., 2012*).

Plexin signaling is critically dependent on the on/off switch of the RapGAP activity under the control of semaphorin (*Wang et al., 2012*). Our previous structural analyses have suggested that the plexin GAP is autoinhibited by adopting a closed conformation that sequesters the active site (*He et al., 2009*). A pre-formed inhibitory dimer of plexin may also be involved in suppressing the GAP activity prior to semaphorin binding (*Antipenko et al., 2003*; *Tong et al., 2007*; *Nogi et al., 2010*). Semaphorins are dimeric molecules and have been suggested to induce dimerization or oligomerization

of plexin for triggering downstream signaling (*Klostermann et al., 1998*; *Koppel and Raper, 1998*; *Driessens et al., 2001*; *Perrot et al., 2002*; *Antipenko et al., 2003*; *Love et al., 2003*). A model of plexin activation involving oligomerization mediated by the RBD/RhoGTPase interaction has been proposed (*Bell et al., 2011*), but existence of this oligomeric structure in solution or on the cell surface has not been established (*Siebold and Jones, 2013*). Recent structural studies have demonstrated how dimeric semaphorin brings two copies of the plexin extracellular region into proximity (*Janssen et al., 2010*; *Liu et al., 2010*; *Nogi et al., 2010*; *Janssen et al., 2012*). We have shown that the purified plexin cytoplasmic region displays low RapGAP activity, which can be activated dramatically by fusing it to the coiled-coil dimerization motif of GCN4 (general control non-repressed 4) (*Wang et al., 2012*). These observations collectively support that semaphorin-induced formation of an active dimer of plexin is the major mechanism for activation of the GAP domain and intracellular signaling.

In this study we sought to understand how the plexin RapGAP is activated by induced-dimerization and facilitates GTP hydrolysis specifically for Rap. We systematically screened various coiled-coil dimer fusions of the plexin cytoplasmic region for optimal activation of the GAP. These experiments led to crystallization and structure determination of the active dimer of zebrafish PlexinC1. In addition, we employed a novel protein ligation system to covalently link the plexin cytoplasmic region and Rap, which stabilized their weak interaction and allowed us to crystallize and determine the structure of a PlexinC1/Rap complex. The structures and the associated mutational analyses together reveal the basis for the dimerization-induced activation, the non-canonical catalysis and the unique specificity of plexin for Rap.

## Results and discussion

### Screening and crystallization of the coiled-coil-induced active dimer of plexin_{cyto}

Our previous study has shown that the RapGAP activity of the cytoplasmic region of plexins (plexins_{cyto}) can be activated by fusing it to the coiled-coil motif of GCN4 through a flexible linker of various lengths (*Wang et al., 2012*). Our extensive crystallization trials of these coiled-coil induced dimers of plexins_{cyto} all failed, presumably due to the flexibility of the linker. We therefore removed the linker and directly fused the coiled-coil with the juxtamembrane helix (the N-terminal helix in the juxtamembrane segment) of plexin_{cyto} (*Figure 1*). Assuming the coiled-coil motif and the juxtamembrane helix of plexin merge into a continuous helix, varying the relative register between them by adding or removing residues at the junction can result in dramatically different relative orientations between the two plexin monomers in the dimer. Without knowing the ideal arrangement of the two monomers for active dimer formation, we systematically tested fusing plexin_{cyto} to each of the seven unique positions on the heptad repeat of the coiled-coil (*Figure 1A*). We used mouse PlexinA1_{cyto} for the screening experiments, because it displayed the highest level of activation by induced dimerization in our previous study (*Wang et al., 2012*). We chose Ala1272, located near the N-terminus of the juxtamembrane helix in PlexinA1_{cyto}, as the reference for naming the fusion constructs. These constructs are referred to as CC(x)PlexinA1_{cyto}, in which 'x' indicates the position of Ala1272 in PlexinA1 on the heptad repeat (*Figure 1A*).

GAP activity assays showed that all these dimer constructs are substantially more active than the monomer (*Figure 1C*). Remarkably, CC(a)PlexinA1_{cyto}, CC(d)PlexinA1_{cyto} and CC(g)PlexinA1_{cyto}, which confer in general similar inter-monomer orientations, achieve much higher activation levels than CC(b)PlexinA1_{cyto}, CC(c)PlexinA1_{cyto}, CC(e)PlexinA1_{cyto} and CC(f)PlexinA1_{cyto}. We also tested four zebrafish CC(x)PlexinC1_{cyto} constructs, which showed the same trend of activation levels (*Figure 1A,D*). These results further support the notion that a specific association mode between the two plexin monomers is required for the optimal dimerization-induced activation (*Wang et al., 2012*). We screened for crystals of those highly active dimer constructs, and obtained crystals of zebrafish CC(a)PlexinC1_{cyto} and determined the structure at 3.3 Å resolution (*Table 1*).

### Characterization and crystallization of ligated plexin_{cyto}/Rap complexes

Our attempts to co-crystallize Rap with various plexins_{cyto} also failed, likely due to their weak interaction (*Wang et al., 2012*). To stabilize the interaction, we covalently linked plexins_{cyto} and Rap1B in vitro by using a protein ligation system based on the transpeptidase activity of sortase from *Staphylococcus aureus* (*Figure 2*; see details in 'Materials and methods') (*Popp et al., 2009*). We used the GAP activity assay to characterize the ligated complex of zebrafish PlexinC1_{cyto} and human Rap1B connected by a

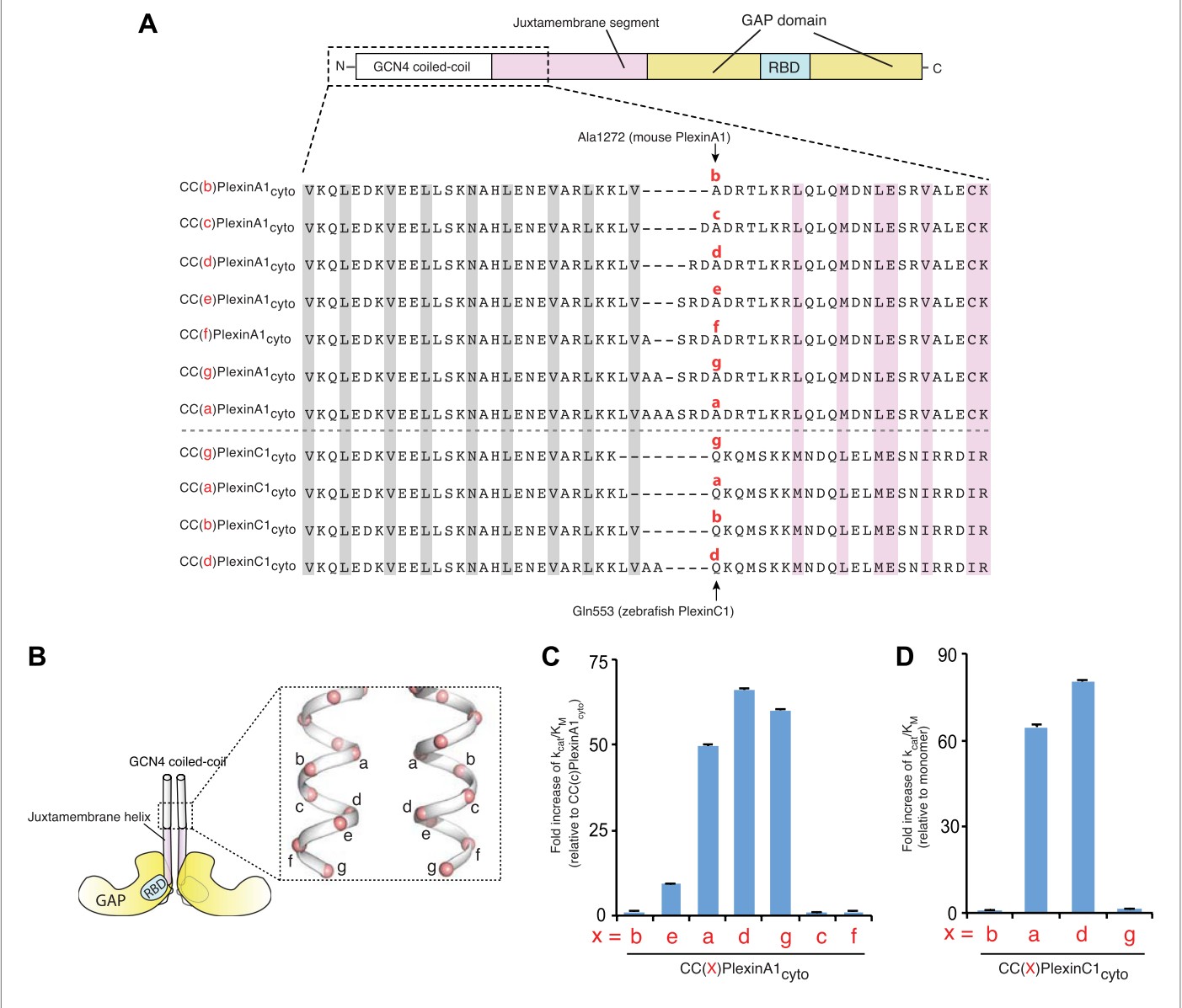

**Figure 1**. Activation of the plexin GAP by coiled-coil fusion. (**A**) Design of the coiled-coil fusions of mouse PlexinA1$_{cyto}$ and zebrafish PlexinC1$_{cyto}$. The juxtamembrane segment sequences from mouse PlexinA1$_{cyto}$ and zebrafish PlexinC1$_{cyto}$ are aligned. The constructs are named CC(x)Plexin$_{cyto}$, where 'x' (in red) is the position of Ala1272 in PlexinA1 or Gln553 in PlexinC1 on the heptad repeat. The 'a' and 'd' positions in the GCN4 coiled-coil are highlighted gray. Residues at the active dimer interface are highlighted pink. (**B**) Diagram of the CC(x)Plexin$_{cyto}$ constructs. (**C**) GAP activity of mouse CC(x)PlexinA1$_{cyto}$. Activity of monomeric PlexinA1$_{cyto}$ is too low to be measured reliably. The fold increase of $k_{cat}/K_M$ is calculated relative to CC(c) PlexinA1$_{cyto}$, which is the least active among the dimers but approximately 10-fold more active than the monomer. (**D**) GAP activity of zebrafish CC(x) PlexinC1$_{cyto}$. In both (**C**) and (**D**), error bars represent standard error of the $k_{cat}/K_M$.

24-residue flexible linker and the sortase-recognition motif. The ligated complex catalyzes GTP hydrolysis at much higher rates than the two individual proteins mixed at the same concentrations (*Figure 2C*), indicating enhanced formation of the catalytically competent plexin/Rap complex when the two proteins are tethered.

As mentioned above, the dimerization induces the active conformation of plexin$_{cyto}$, which enhances Rap binding and GTP hydrolysis. Conversely, stabilization of the active conformation of plexin by Rap binding is expected to facilitate formation of the plexin dimer. Due to basal GAP activity of plexin$_{cyto}$, Rap in the ligated plexin$_{cyto}$/Rap complex is GDP-bound and cannot stably

**Table 1.** Data collection and refinement statistics

| Data collection | | |
| --- | --- | --- |
| Crystal | CC(a)PlexinC1$_{cyto}$ | PlexinC1$_{cyto}$/Rap1B |
| Space group | P2$_1$2$_1$2$_1$ | P1 |
| Cell dimensions | | |
| a, b, c (Å) | 53.22, 146.10, 209.58 | 76.28, 84.73, 138.75 |
| α, β, γ (°) | 90, 90, 90 | 91.09, 95.15, 90.32 |
| Resolution (Å) | 50.0–3.30 (3.36–3.30)* | 50.0–3.30 (3.36–3.30)* |
| $R_{sym}$ | 11.1(86.8) | 5.2(48.7) |
| I/σ | 19.7(1.4) | 18.6(1.6) |
| Completeness (%) | 95.8(79.5) | 91.0(89.2) |
| Redundancy | 10.4(4.3) | 1.9(1.9) |
| **Refinement** | | |
| Resolution (Å) | 3.30 | 3.30 |
| No. reflections | 21,087 | 47,207 |
| Completeness(%) | 83.32† | 90.23 |
| $R_{work}$/$R_{free}$ (%) | 22.6/28.2 | 24.3/30.0 |
| No. atoms | 8888 | 22,228 |
| Protein | 8871 | 22,086 |
| Ligand/ion | 0 | 132 |
| Water | 17 | 10 |
| B-factors | | |
| Protein | 98.9 | 143.5 |
| Ligand/ion | – | 128.2 |
| Water | 49.3 | 89.7 |
| R.m.s deviations | | |
| Bond lengths (Å) | 0.005 | 0.004 |
| Bond angles (°) | 0.85 | 0.70 |
| Ramanchandran plot | | |
| Favored (%) | 91.7 | 93.1 |
| Allowed (%) | 8.1 | 6.7 |
| Disallowed (%) | 0.2 | 0.2 |

*Highest resolution shell is shown in parenthesis.
†The data were corrected for anisotropy in HKL2000. This treatment eliminated many weak reflections and reduced the completeness of the data used for refinement compared to the completeness reported for data collection.

bind or induce dimerization of plexin$_{cyto}$. We therefore used the γ-phosphate analog aluminum fluoride (AlF$_x$, x = 3 or 4) (***Vetter and Wittinghofer, 2001***) to induce formation of the transition state complex between Rap(GDP) and plexin. Our analytical ultracentrifugation experiments showed that while the ligated PlexinC1$_{cyto}$/Rap1B complex itself did not dimerize, it dimerized robustly in the presence of AlF$_x$ (***Figure 2D***). We crystallized this complex with AlF$_x$ and determined the structure to 3.3 Å resolution (***Table 1***).

## Overall structures of the CC(a) PlexinC1$_{cyto}$ dimer and the PlexinC1$_{cyto}$/Rap complex

In the CC(a)PlexinC1$_{cyto}$ structure, the two plexin monomers in the asymmetric unit form a symmetric side-by-side dimer (***Figure 3A***). The two juxtamembrane helices are oriented approximately in parallel, extending well beyond the main body of the proteins and integrating into the C-termini of the coiled-coil moiety. On the plasma membrane, this configuration of the plexin dimer orients the active sites of the two GAP domains toward the membrane surface and leaves sufficient space for binding of the membrane anchored Rap substrate, as observed in the PlexinC1$_{cyto}$/Rap1B complex structure (***Figure 3A,B***).

The asymmetric unit of the PlexinC1$_{cyto}$/Rap1B complex structure contains four protomers of the complex, which are virtually identical to one another. The four PlexinC1 molecules form two pairs of dimers, consistent with the dimerization observed in solution. The conformation of PlexinC1 and its mode of dimerization are highly similar in the two structures (***Figure 3C***), supporting that they represent the active state of plexin and are not artifacts induced by the fusion constructs.

The coiled-coil moiety in the CC(a)PlexinC1$_{cyto}$ structure is nearly identical to the isolated coiled-coil structures reported previously (***O'Shea et al., 1991***). Comparison of the active dimers in the two structures suggests that there is a small geometric incompatibility between the coiled-coil and the plexin dimer, as the N-terminal portion of the juxtamembrane helix (residues 553–566) seems to bend slightly near its junction with the coiled-coil (***Figure 3C***). This portion of the juxtamembrane helix does not mediate any inter-molecular interactions and likely has some flexibility. The flexibility can further compensate for deletion or insertion of one residue at the junction between the coiled-coil and the juxtamembrane helix, allowing several constructs to induce the active dimer and achieve similarly high activation levels (***Figure 1***). More deletions or insertions at the junction probably cannot be accommodated without severe distortion of the juxtamembrane helix, explaining the much lower activation levels of those constructs (***Figure 1***).

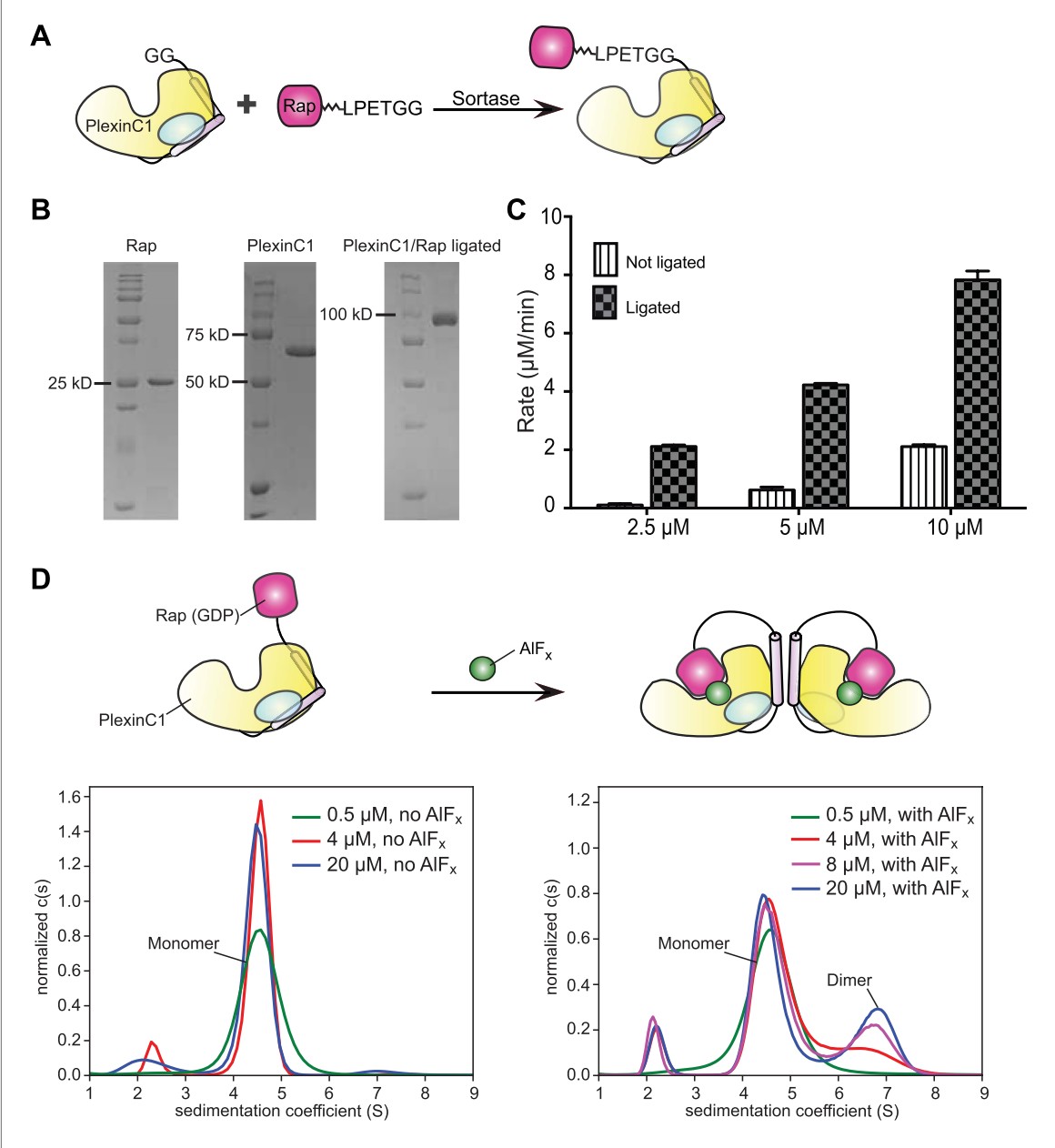

**Figure 2**. Sortase-mediated ligation and characterization of the ligated PlexinC1$_{cyto}$/Rap1B complex. (**A**) Scheme of the sortase-mediated ligation. (**B**) Representative gels of purified PlexinC1$_{cyto}$, Rap1B and the ligated PlexinC1$_{cyto}$/Rap1B complex with the 24-residue linker and the 'LPETGG' sortase recognition motif. (**C**) Comparison of the GTP hydrolysis activity between the ligated complex and the individual PlexinC1$_{cyto}$ and Rap1B proteins mixed at the same concentrations. The hydrolysis rates are averages of three replicates. Error bars represent standard deviation of the mean. (**D**) Analytical ultracentrifugation showing AlF$_x$-induced dimerization of the ligated PlexinC1$_{cyto}$/Rap1B complex. In the absence of AlF$_x$ (the left panel), the majority of the complex behaves as a monomer with a sedimentation coefficient of 4.5 S. In the presence of AlF$_x$ (the right panel), a dimeric species (sedimentation coefficient of 6.7 S) appears and becomes more abundant at higher protein concentrations.

## Interactions in the dimer interface

We will refer to the CC(a)PlexinC1$_{cyto}$ structure for the following discussion on the active dimer unless otherwise stated, because the dimer interface in this structure is better resolved in the electron density map. The dimer interface is formed by the juxtamembrane helix and one side of the GAP domain, burying a total of ~3200 Å² surface area (**Figure 4**). The RBDs in the two monomers are far away from each other and not involved in dimer formation. The center of the dimer interface is a 4-helix bundle

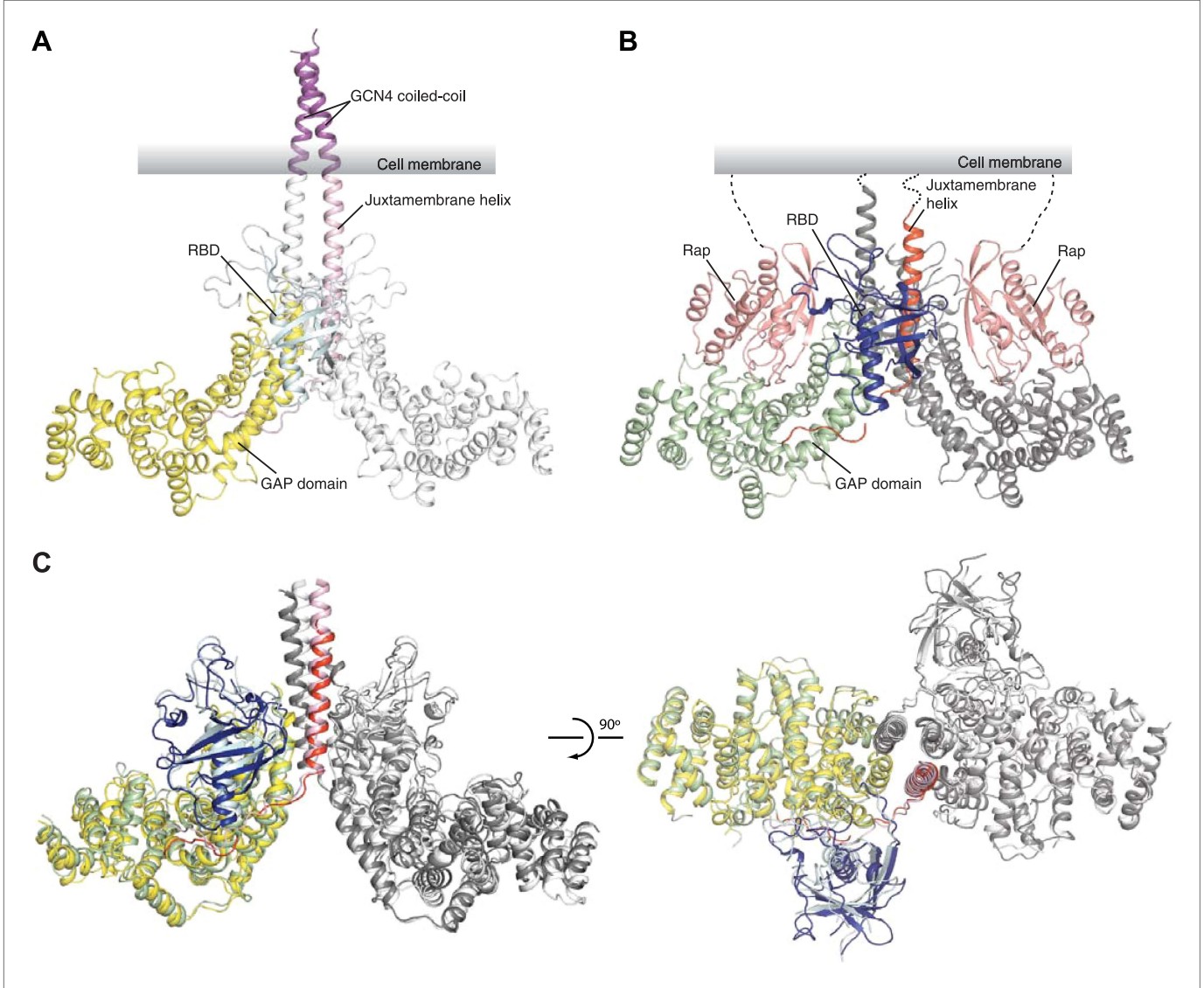

**Figure 3**. Overall structures of the zebrafish CC(a)PlexinC1$_{cyto}$ active dimer and the PlexinC1$_{cyto}$/Rap1B complex. (**A**) Structure of the CC(a)PlexinC1$_{cyto}$ dimer. (**B**) Structure of the PlexinC1$_{cyto}$/Rap1B complex. One of the two active dimers of plexin with Rap1B bound in the asymmetric unit is shown. In both (**A**) and (**B**), domains from one plexin monomer in the dimer are colored and labeled. The other monomer is shown in white in (**A**) and gray in (**B**). (**C**) Comparison of the active dimers in the structures of CC(a)PlexinC1$_{cyto}$ and the PlexinC1$_{cyto}$/Rap1B complex. The coiled-coil moiety is omitted for clarity. The color schemes are the same as in (**A**) and (**B**).

structure comprised of the C-terminal portion of the juxtamembrane helix (residues 567–584) and the N-terminal portion of helix 11 in the GAP domain (residues 929–943) from each monomer (*Figure 4*). The core of the 4-helix bundle is dominated by hydrophobic interactions, involving residues Ile571, Ile575, Phe579 and Leu582 from the juxtamembrane helix and Met933, Ile936 and Leu939 from helix 11 (*Figure 4B*). The core interface is supported by peripheral electrostatic interactions mediated by Arg572, Arg576 and Asp581 from the juxtamembrane helix and Glu770, Glu932, and Lys937 from the GAP domain (*Figure 4A*).

A loop-helix segment (residues 1038–1058) between helix 15 and 17 in the GAP domain of each monomer wraps around the C-terminal portion of the 4-helix bundle. The interactions involve Leu1045, Lys1047, Leu1054, Leu1055, and Lys1058 in the loop-helix segment and Phe579, Gln583, Thr584 and Leu939 from the 4-helix bundle (*Figure 4A*). We call the loop-helix element 'the activation segment' in plexin since it plays a major role in regulating the GAP activity (see the next section for details), functionally resembling the well known activation segment in protein kinases (*Huse and Kuriyan, 2002*).

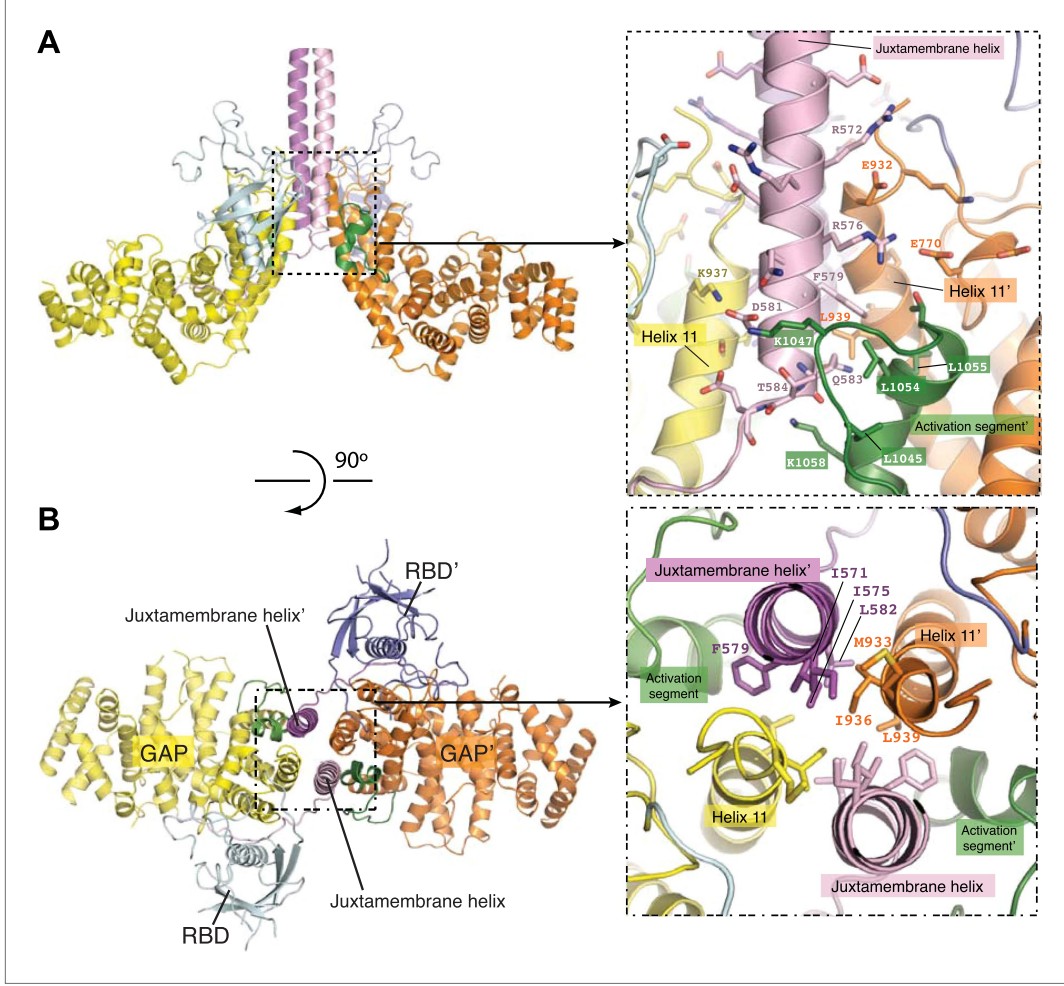

**Figure 4**. The dimer interface in the CC(a)PlexinC1$_{cyto}$ structure. (**A**) Periphery of the dimer interface. The coiled-coil moiety is not shown. (**B**) Hydrophobic core of the dimer interface. Residue labels for one monomer are omitted for clarity.

## Dimerization induced conformational changes that lead to GAP activation

Comparisons of the dimer structure with previously determined structures of plexins$_{cyto}$ reveal several substantial conformational differences. The most striking difference is in the juxtamembrane helix (**Figure 5A**). Except in one of the PlexinB1 structures where it is disordered (**Bell et al., 2011**), the juxtamembrane helix in all other previous structures adopts a kinked conformation, with both the N- and C-terminal halves interacting with the GAP domain (**He et al., 2009**; **Tong et al., 2009**; **Wang et al., 2012**). In the active dimer structure, the last two turns in the juxtamembrane helix (residues 585–591, corresponding to residues 1282–1288 in mouse PlexinA3) convert to an extended loop. This loop and the following segment use Asp588, Leu589, Asp591 and Val593 to make a distinct set of intra-molecular interactions with the GAP domain (**Figure 5B**). The remaining N-terminal helical portion (residues 553–584) adopts a straight conformation and rotates by ~90° in relation to the inactive structures (**Figure 5A**) to mediate the formation of the 4-helix bundle at the center of the dimer interface (**Figure 4B**). Helix 11 undergoes a small tilt to accommodate the juxtamembrane helix from the dimer partner, and the top part (residues 929–934) adopts a 3$_{10}$ helix like conformation to pack against the hydrophobic core of the 4-helix bundle (**Figure 5C**).

The conformational changes in the juxtamembrane helix and helix 11 are coupled to changes in the activation segment. In all the previously reported structures of plexins$_{cyto}$, the highly conserved helical portion of the activation segment adopts essentially the same 'closed' conformation (**Figure 6A**) (**He et al., 2009**; **Tong et al., 2009**; **Bell et al., 2011**; **Wang et al., 2012**). An asparagine residue in the

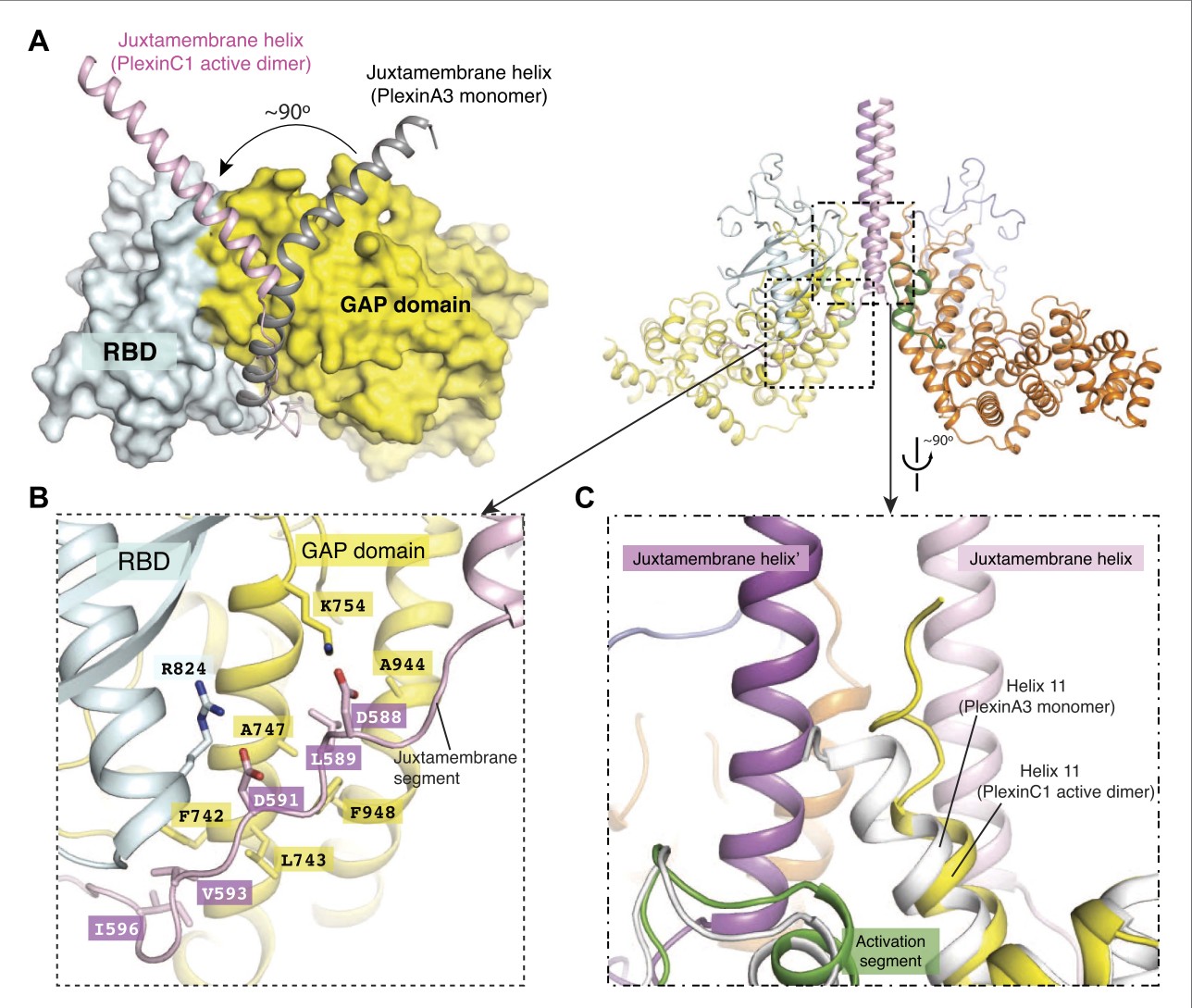

**Figure 5**. Dimerization-induced conformational changes of the juxtamembrane helix and helix 11. (**A**) Conformational change of the juxtamembrane helix. One monomer in the PlexinC1$_{cyto}$ active dimer is superimposed onto the monomeric PlexinA3$_{cyto}$ structure (PDB code: 3IG3). The GAP domain and RBD of PlexinA3$_{cyto}$ are shown in the surface representation. (**B**) Intra-molecular interactions made by the extended portion of the juxtamembrane segment in the CC(a)PlexinC1$_{cyto}$ structure. (**C**) Conformational change of helix 11. The structure superimposition is the same as in (**A**).

helix (Asn1774 in mouse PlexinA3) is invariably hydrogen bonded with a conserved aspartate (Asp1758 in PlexinA3) in helix 15. A proline residue (Pro1772 in PlexinA3) at the N-terminus of the helix acts as a lid that covers the asparagine and blocks its access to the incoming Rap substrate. Docking Rap to PlexinA3 based on the PlexinC1/Rap complex structure results in a number of clashes between Rap and the activation segment (***Figure 6A***). The proline 'lid' (Pro1772) sterically clashes with Tyr40 in Rap, while the carbonyl oxygen on the sidechain of Asn1774 makes an unfavorable contact with the sidechain of Asp38 in Rap. The loop portion of the activation segment appears to be rather flexible, as it displays high B-factors in PlexinA3 (PDB ID: 3IG3) and the PlexinB1/Rac1 complex (PDB ID: 3SU8) and is partially disordered in apo-PlexinB1 (PDB ID: 3HM6) and the PlexinA1/Rac1 complex (PDB ID: 3RYT). The loop likely samples many conformations, some of which may impose additional hindrance on Rap binding.

In contrast, the activation segment in the active dimer adopts an open conformation and shifts away from the GAP active site (***Figure 6B,C***). This shift appears to be induced by the interactions between the activation segment and the 4-helix bundle in the dimer interface (***Figure 6C***). The outward

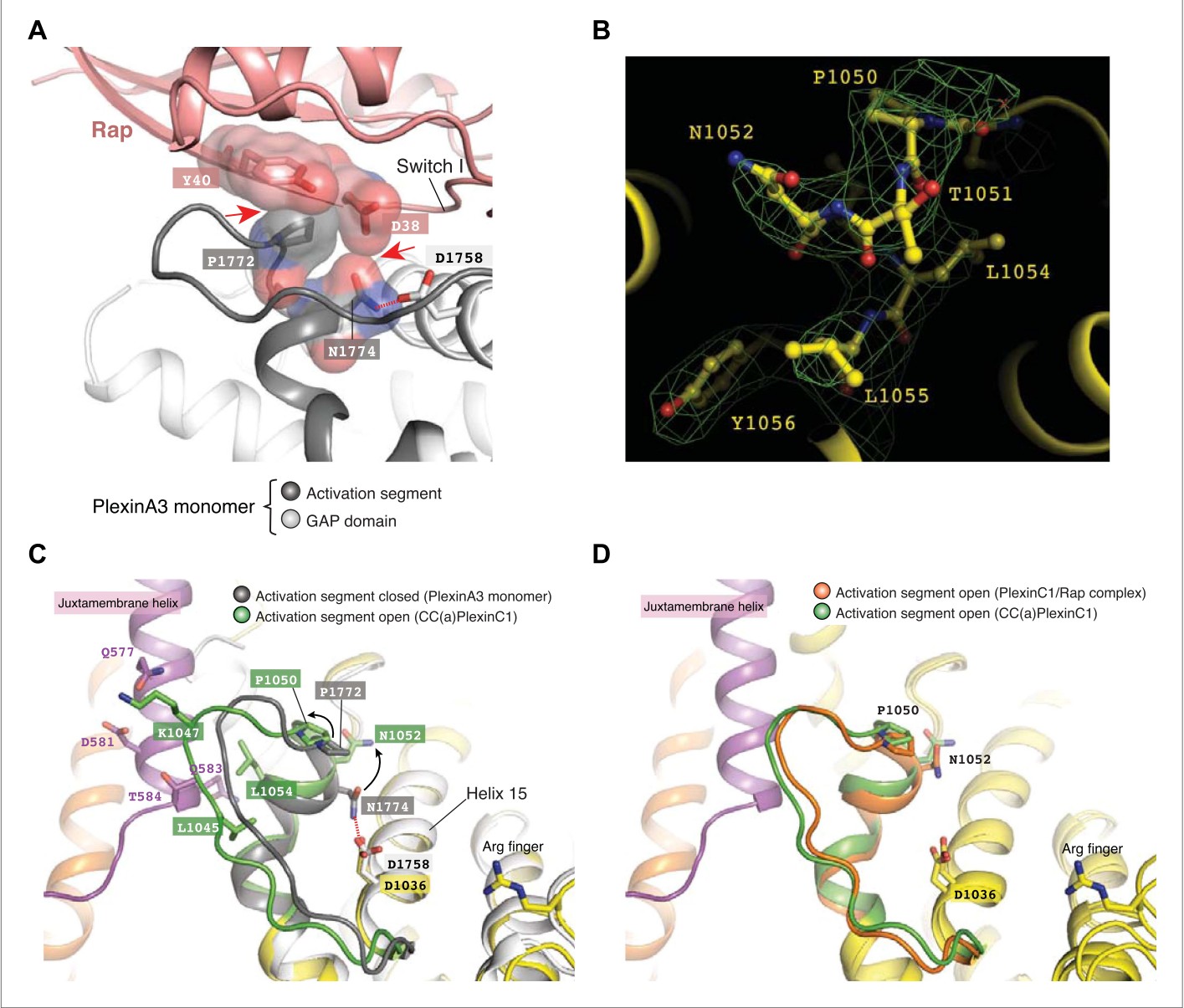

**Figure 6**. Dimerization-induced opening of the activation segment. (**A**) Docking of Rap to the inactive PlexinA3$_{cyto}$ structure (PDB code: 3IG3). The docking is based on a superimposition between PlexinA3 and PlexinC1 in the PlexinC1/Rap complex structure (see 'Materials and methods' for details). Red dashed line: hydrogen bond. Red arrows: steric clashes and unfavorable interactions. (**B**) Sigma-A weighted simulated annealing omit map of the activation segment in CC(a)PlexinC1$_{cyto}$. The map was calculated using the model with residues 1050–1056 in one of PlexinC1 molecules removed. The map was contoured at 3σ, with the final model shown. (**C**) Comparison of the activation segment in the structures of CC(a)PlexinC1$_{cyto}$ and PlexinA3$_{cyto}$. Conformational differences important for GAP activation are highlighted by black arrows. (**D**) Comparison of the activation segment in the structures of CC(a)PlexinC1$_{cyto}$ and the PlexinC1$_{cyto}$/Rap complex.

shift pulls Asn1052 (Asn1774 in PlexinA3) away from Asp1036 (Asp1758 in PlexinA3), precluding hydrogen bond formation. Pro1050 in the dimer structure also moves outward compared to Pro1772 in PlexinA3 (*Figure 6C*). The activation segment in the structure of the PlexinC1/Rap complex adopts a similar open conformation (*Figure 6D*). Therefore, a major mechanism in the dimerization-induced activation of plexin appears to be the outward shift of the activation segment, which opens the otherwise obstructed active site to allow Rap binding and catalysis of GTP hydrolysis. While this conformational change in the plexin GAP domain seems small, it is known that interactions between small GTPases and their regulators or effectors can be strongly influenced by subtle changes at the binding interface

(*Nassar et al., 1996*; *Snyder et al., 2002*). The activation segment in the PlexinC1/Rap complex is slightly more closed than that in the coiled-coil-induced PlexinC1 dimer (*Figure 6D*), indicating that the active dimer promotes a conformation that is more open than required for accommodating Rap. Binding of Rap induces a slight closure of the active site for optimal interactions and catalysis of GTP hydrolysis.

The RBD and the subdomain composed of the first three and the last two helices in the GAP domain show conformational variations among all the structures of plexins. Given the fact that they are not involved in the dimer interface or Rap binding, the variations of these structural elements likely reflect their intrinsic flexibility and are not relevant to the activation mechanism.

## Mutational analysis of the active dimer structure

We performed extensive mutational analyses to test the activation mechanism revealed by the dimer structure. Arg576, Asp581, Asp588, Val593 and Met933 are involved in the dimer interface or intra-molecular interactions that stabilize the new conformation of the juxtamembrane segment (*Figures 4 and 5B*). In the inactive monomer structures, residues at these positions are surface exposed and do not make any interactions. We made the R576E, D581K, D588K, V593E and M933E single mutations in CC(a)PlexinC1$_{cyto}$. The GAP assay showed that R576E, D581K, D588K and M933E strongly impaired dimerization-induced activation (*Figure 7A*). The deleterious effect of V593E on GAP activation is weaker but clearly observable at a lower plexin concentration (*Figure 7A,C*). To test the coupling between the dimerization and the conformation of the activation segment, we designed the Q583A, T584A, L1045A, K1047A and L1054A mutations to disrupt the interactions between the activation segment and the juxtamembrane helix from the dimer partner (*Figure 6C*). The GAP assay showed that while K1047A modestly decreased dimerization-induced GAP activation, L1054A, L1045A and T584A greatly reduced the activation (*Figure 7B,C*).

We further examined the activation mechanism by using a functional assay, which assesses the ability of plexin to induce COS7 cell collapse upon semaphorin stimulation (*Takahashi et al., 1999*). Since the ligand for zebrafish PlexinC1 was not available, mouse PlexinA3 and its ligand Sema3F were used in these assays (*He et al., 2009*). The K1273E, E1278R, E1285R and M1290E mutations of mouse PlexinA3, corresponding to R576E, D581K, D588K and V593E of zebrafish PlexinC1 respectively, all significantly impaired plexin-mediated COS7 cell collapse (*Figure 7D*). A previous study identified a large panel of mutations that abolished PlexinB1-mediated COS7 cell collapse (*Bell et al., 2011*). These mutations were designed to test the model of plexin activation by Rac1-induced oligomerization. The results are also consistent with the activation mechanism shown here, as most of the mutated resides are conserved in zebrafish PlexinC1 and are involved in formation of the active dimer. Some mutations of highly conserved residues in the dimer interface have been identified in cancer patients, including R2040W in PlexinB1 (*Gui et al., 2011*) (corresponding to Lys1058 in zebrafish PlexinC1) and R1680Q/W in PlexinA2 (*Cancer Genome Atlas Research Network, 2012*) (corresponding to Lys937 in zebrafish PlexinC1). Both of these mutations likely prevent formation of the active dimer of plexins, consistent with the tumor suppressor function of plexins suggested by previous studies (*Gu and Giraudo, 2013*).

## Overall binding mode between zebrafish PlexinC1$_{cyto}$ and human Rap1B

The species mismatch of the plexin/Rap complex does not affect their interaction, since the human Rap1B construct contains only three residues non-identical to their counterparts in zebrafish Rap1B, which are all located far from the plexin/Rap interface (*Figure 8A*, middle panel). The linker between PlexinC1$_{cyto}$ and Rap1B is not visible in the electron density map, suggesting that it is flexible as designed and does not impose restraints on the plexin/Rap interaction. A superimposition of Rap and Ras in the PlexinC1$_{cyto}$/Rap and p120GAP/Ras complexes shows that the overall binding modes of the two with their respective GAPs are similar (*Figure 8A*, left panel) (*Scheffzek et al., 1997*). The GAP domain in PlexinC1 and Switches I (residues 30–38) and II (residues 59–67) in Rap constitute the majority of the binding interface, whereas the RBD in plexin is not involved and its role in GAP regulation remains unclear (*Figure 8*). The core of the interface is composed of several hydrophobic residues, which are surrounded by numerous charge–charge interactions at the periphery. Most of the Rap-binding residues are conserved among the plexin family members, suggesting that they all interact with Rap in the same mode (*Figure 8B,C*). The presumed arginine finger (Arg711) in PlexinC1 superimposes well with the arginine finger (Arg789) in p120GAP, playing the same role in catalysis by interacting with the AlF$_x$ and GDP in the active site (*Figure 8A*). While the bound AlF$_x$ is not clearly resolved in the relatively

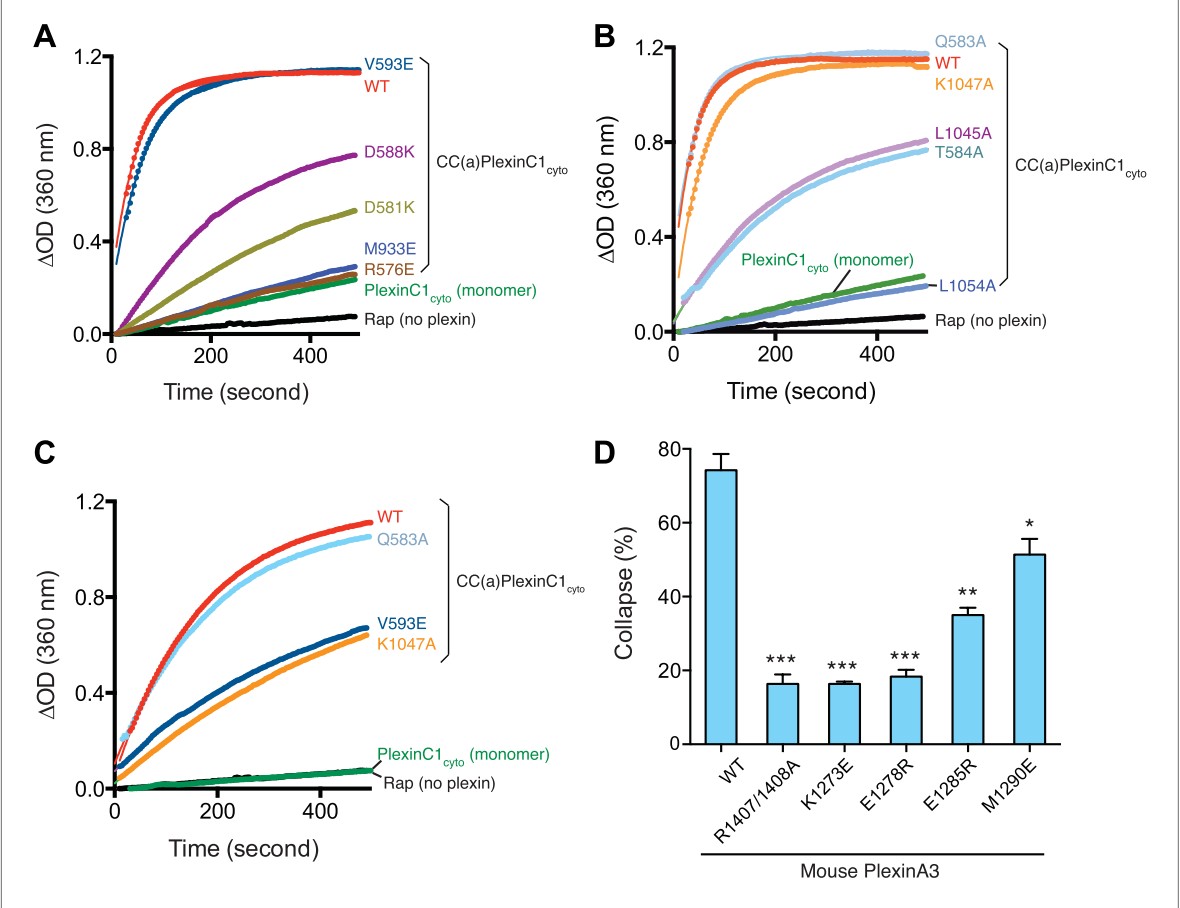

**Figure 7**. Mutational analyses of the dimerization-driven activation mechanism. (**A**–**C**) Mutational analyses of the activation mechanism using the GAP activity assay. Residues mutated in (**A**) are involved in stabilizing the active dimer, whereas residues in (**B**) couple the dimer formation to the opening of the activation segment. In (**A**) and (**B**) the concentration of plexin is 2 μM. In (**C**), the concentration of plexin is 0.25 μM. The Rap concentration is 120 μM for all the assays. Data shown are representative of three replicates. (**D**) Mutational analyses using the COS7 cell collapse assay. The results for the wild type and the arginine-finger mutant (R1407/1408A) are shown as positive and negative controls, respectively. Error bars represent standard error of the mean from three independent experiments. At least 150 cells were counted for each sample in each experiment. Statistical significance between wild type and each mutant is determined by two-tailed Student's $t$-test (\*p<0.05; \*\*p<0.01; \*\*\*p<0.001).

low-resolution map, the shape of the density suggests that it is the trigonal $AlF_3$, the same as in the p120GAP/Ras structure. We therefore modeled it $AlF_3$ in the structure. The second conserved arginine (Arg1001) in PlexinC1 is equivalent to Arg903 in p120GAP, which stabilizes the position of the arginine finger (**Figure 8A**). The functional importance of these two arginine residues in plexin has been demonstrated by previous mutational studies (**Rohm et al., 2000**; **Oinuma et al., 2004**; **He et al., 2009**; **Wang et al., 2012**).

## Interaction between the activation segment in plexin and the Switch I of Rap

Switch I of Rap makes numerous interactions with the activation segment in PlexinC1 (**Figure 9A**). As mentioned above, the activation segment in the PlexinC1/Rap complex structure adopts the open conformation similar to that in the CC(a)PlexinC1 dimer structure. Pro1050 at the N-terminus of the helical portion of the activation segment packs against Tyr40 in Rap. Asn1052 forms two hydrogen bonds with the carboxyl group of Asp38 and the backbone amide of Ser39 in Rap. Lys1053 apparently makes electrostatic interactions with Asp38 in Rap and Asp1036 in plexin. Gln1032 in helix 15 also contributes to Switch I binding through forming three hydrogen bonds. GAP activity assays showed that while the P1050A mutation caused a modest activity decrease, the Q1032E, N1052E and K1053A mutations largely abolished the activity (**Figure 9B**). COS7 cell collapse assays showed that both the

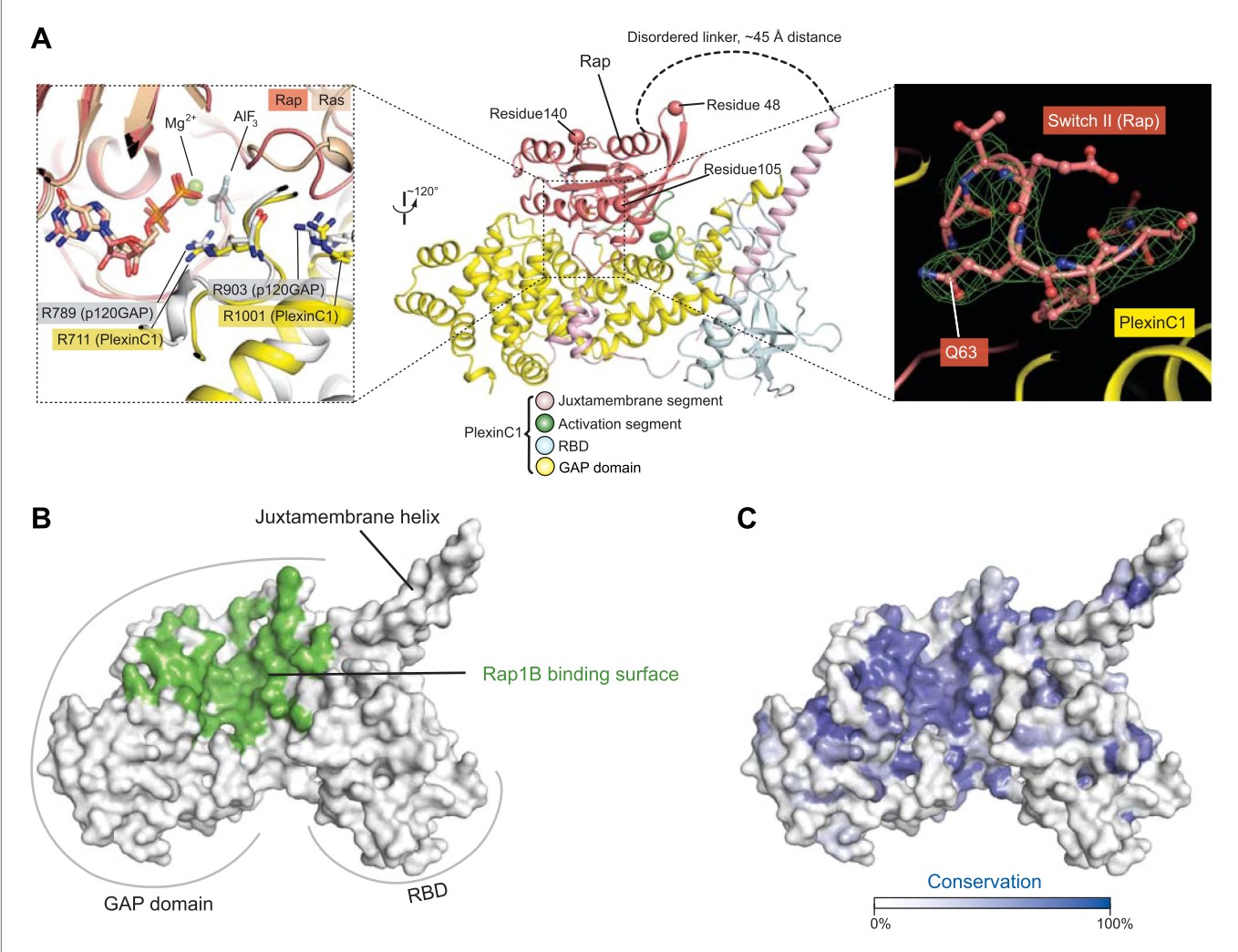

**Figure 8**. Overall view of the interface between zebrafish PlexinC1$_{cyto}$ and human Rap1B in the complex structure. (**A**) The PlexinC1$_{cyto}$/Rap interface and its comparison with that in the p120GAP/Ras complex structure. The middle panel shows the overall structure of the PlexinC1$_{cyto}$/Rap complex, with the three residues (48, 105 and 140) different between human and zebrafish Rap1B highlighted. The left panel shows a superimposition of the active sites in the PlexinC1$_{cyto}$/Rap1B and p120GAP/Ras (PDB ID: 1WQ1) structures. The superimposition is based on Rap1B and Ras. The right panel shows a Sigma-A weighted simulated annealing omit map of Switch II in Rap, calculated using the model with residues 60–66 in one of the Rap1B molecules removed. The map is contoured at 3σ, with the final model of the structure shown. (**B**) Rap-binding surface on PlexinC1$_{cyto}$. Residues in PlexinC1 within 4 Å distance of the bound Rap1B molecule are colored green. (**C**) Sequence conservation projection on the molecular surface of PlexinC1$_{cyto}$. The conservation scores were calculated based on an alignment of zebrafish PlexinC1 and all the plexins from mouse (Plexin A1, A2, A3, A4, B1, B2, B3, C1 and D1).

Q1754E and K1775A mutations of mouse PlexinA3, equivalent to zebrafish PlexinC1 Q1032E and K1053A respectively, greatly impaired the cell collapse activity (*Figure 9C*). Mutations of Pro2032 in PlexinB1 and Lys1809 in PlexinB3, equivalent to Pro1050 and Lys1053 in zebrafish PlexinC1 respectively, have also been found in cancer patients (*Cancer Genome Atlas Research Network, 2012*; *Seshagiri et al., 2012*).

## Plexin induces the 'Gln63-in' conformation of Rap for catalysis

Switch II of Rap in the complex structure adopts an unprecedented conformation that is markedly different from both the p120GAP/Ras and the RapGAP/Rap complexes (*Figure 10*) (*Scheffzek et al., 1997*; *Scrima et al., 2008*). Residues 60–63 in Switch II form a tight hairpin-like turn, which brings Gln63 close to AlF₃ (therefore named the Gln63-in conformation). The Gln63 sidechain is placed in a nearly identical position in the active site as Gln61 in the p120GAP/Ras complex (*Figure 10B*).

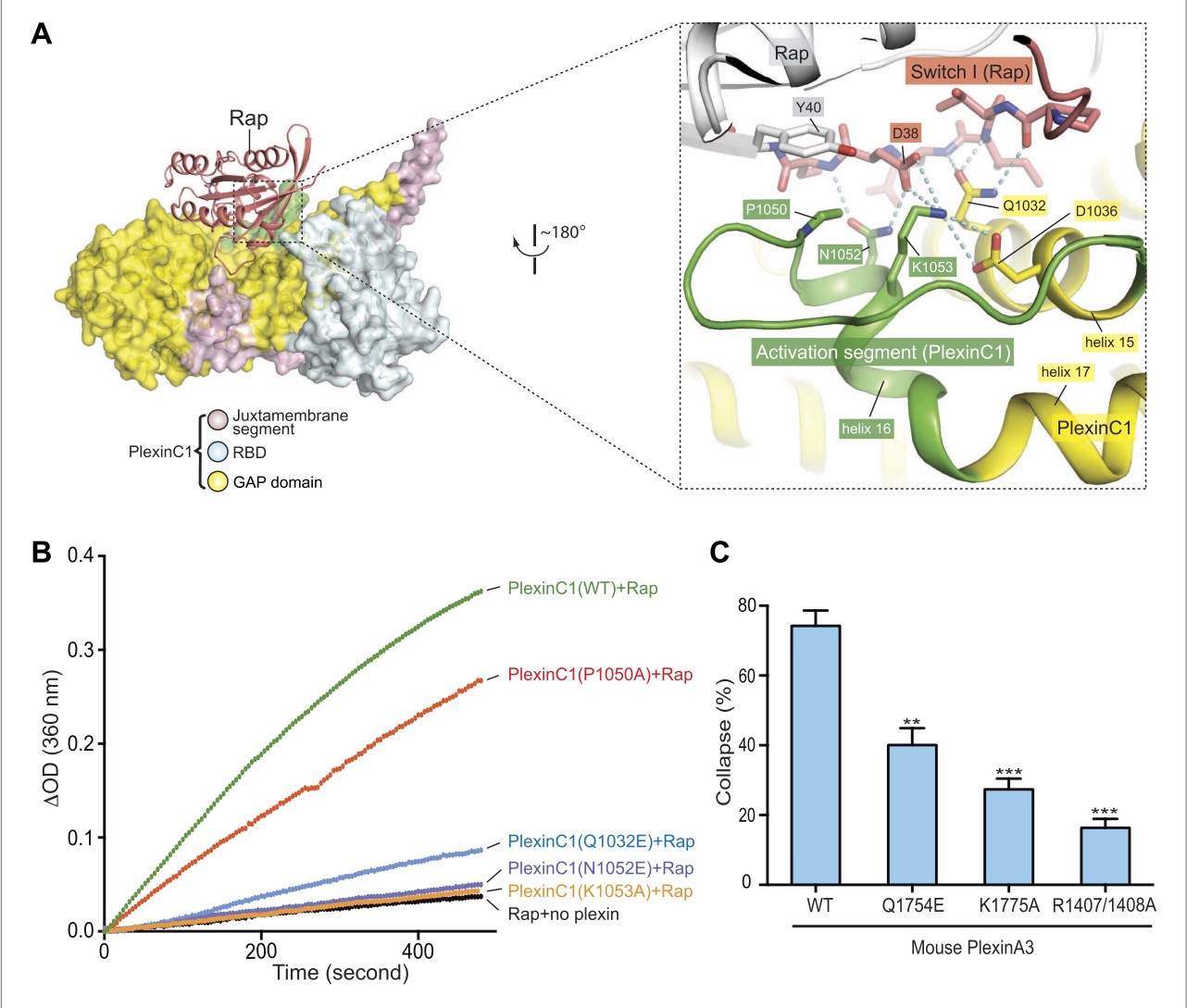

**Figure 9**. Interaction between the activation segment in PlexinC1 and Switch I in Rap. (**A**) Interface between the activation segment and Switch I. Polar interactions and potential hydrogen bonds are indicated by dashed lines. (**B**) GAP activity assays for mutations at the activation segment/Switch I interface. Monomeric PlexinC1$_{cyto}$ was used in these assays. The plots are representatives of three replicates. (**C**) COS7 cell collapse assays for mutations at the activation segment/Switch I interface. Q1754E, and K1775A of mouse PlexinA3 correspond to Q1032E and K1053A of zebrafish PlexinC1, respectively. The data analysis and presentation are the same as in *Figure 7D*.

This comparison strongly supports that Rap Gln63 indeed fulfills the catalytic role of Gln61 in Ras, that is stabilizing the nucleophilic water (*Sot et al., 2010*; *Wang et al., 2012*). Consistently, mutation of Gln63 in Rap has been shown to abolish GTP hydrolysis catalyzed by both plexin and the dual-specificity GAPs (*Sot et al., 2010*; *Wang et al., 2012*). The segment following Gln63 (residues 64–67) adopts an extended conformation, allowing it to span the distance between Gln63 in the active site and the helix following Switch II. In contrast, the corresponding segments in the p120GAP/Ras and the Rap/RapGAP complexes adopt helical structures, holding residue 63 away from the active site (*Figure 10B,C*).

The Gln63-in conformation of Switch II is stabilized by numerous specific interactions between PlexinC1 and Rap. The side chains of Arg1001, Asn1005 and Asn1009 in helices 13 and 14 of PlexinC1 form a network of hydrogen bonds with the backbone of Switch II (*Figure 10D*). Pro611 in the second helix of the juxtamembrane segment makes van der Waals interactions with Thr65 in Switch II (*Figure 10E*). Mutation of either Asn1005 or Asn1009 dramatically decreased the GAP activity (*Figure 10F*).

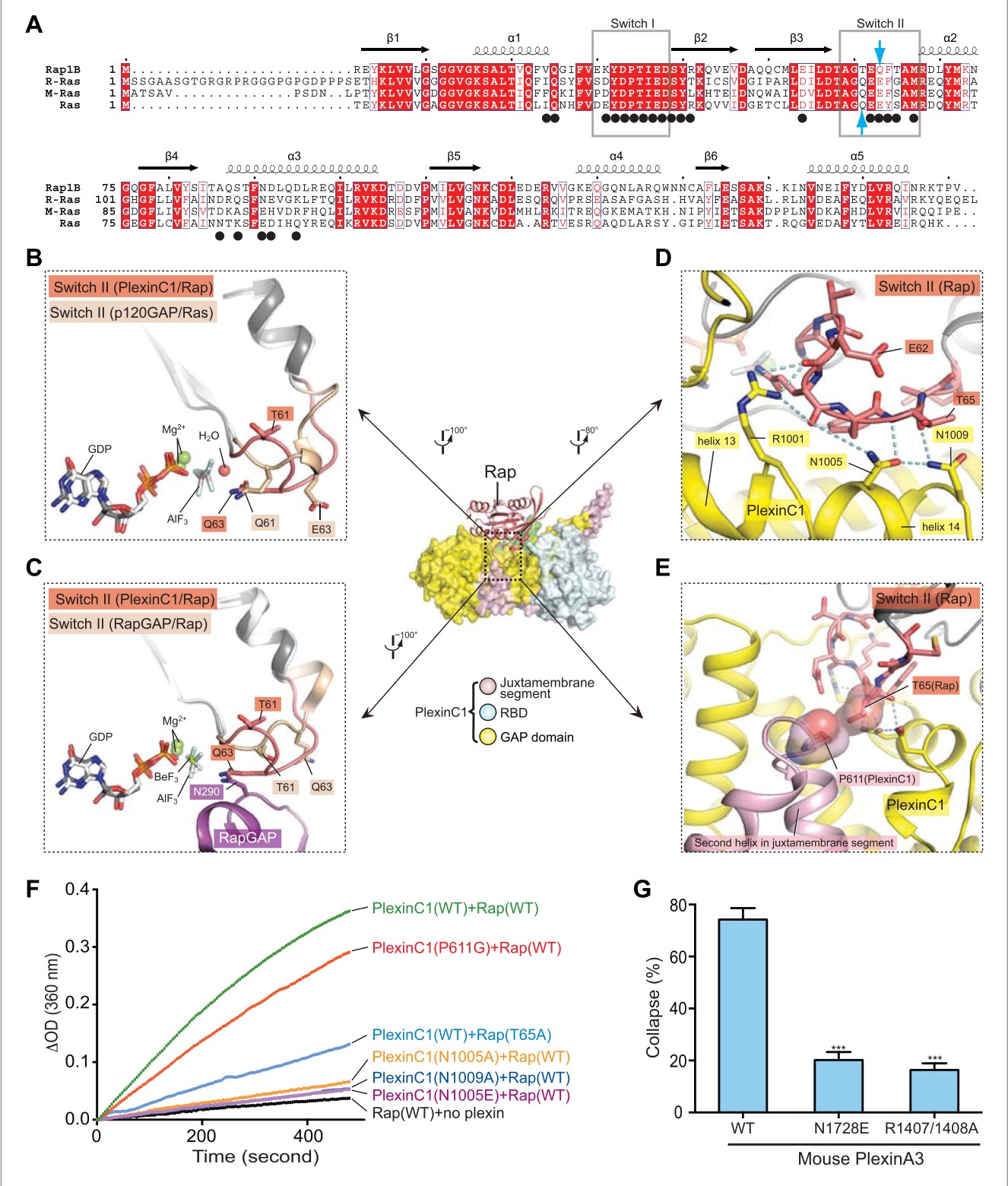

**Figure 10**. The Gln63-in conformation of Switch II in the PlexinC1cyto/Rap1B complex. (**A**) Sequence alignment of human Rap1B, R-Ras, M-Ras and Ras. Black circles denote residues in Rap1B that are involved in binding PlexinC1cyto. Gln63 in Rap1B and Gln61 in Ras are highlighted by blue arrows. (**B**) Comparison of Switch II in the PlexinC1/Rap and the p120GAP/Ras (PDB ID: 1WQ1) complexes. The nucleophilic H₂O is not included in the PlexinC1cyto/ Rap1B structure due to low resolution of the density map. (**C**) Comparison of Switch II in the PlexinC1/Rap and the RapGAP/Rap (PDB ID: 3BRW) complexes.
*Figure 10. Continued on next page*

*Figure 10. Continued*

(**D**) Specific interactions between PlexinC1 and Switch II in Rap1B. Polar interactions and potential hydrogen bonds are indicated by dashed lines. (**E**) Interaction between Pro611 in PlexinC1 and Thr65 in Rap1B. (**F**) GAP activity assays for mutations at the plexin/Switch II interface. Monomeric PlexinC1$_{cyto}$ was used in these assays. The plots are representatives of three replicates. (**G**) COS7 cell collapse assays for mutations at the plexin/Switch II interface. The data analysis and presentation are the same as in *Figure 7D*.

The N1728E mutation in mouse PlexinA3 (equivalent to N1005E of zebrafish PlexinC1) also abolished the cell collapse activity (*Figure 10G*). Mutating Pro611 to glycine, which eliminates its interaction with Thr65 in Switch II, decreased the GAP activity (*Figure 10F*). Conversely, the wild-type PlexinC1 showed decreased activity towards the Rap T65A mutant (*Figure 10F*). The Switch II-interacting residues are highly conserved among the plexin family members, suggesting they all use the same mechanism to stabilize the Gln63-in conformation.

## Specificity determinants in plexin and the dual-specificity GAPs

The dual-specificity GAPs do not share some of the Switch II-interacting residues with plexin (*Figure 11*). For example, Asn1005 in PlexinC1 is replaced by a proline in the dual-specificity GAPs (Pro585 in SynGAP) (*Pena et al., 2008*), lacking the ability to stabilize the Gln63-in conformation of Rap through hydrogen bonds. This loss may be compensated by the extra domains outside of the GAP domain in the dual-specificity GAPs, which have been shown to be required for their RapGAP activity but not for the RasGAP activity (*Kupzig et al., 2006*; *Pena et al., 2008*; *Kupzig et al., 2009*; *Sot et al., 2010*). It has been suggested that the extra domains contribute to the catalysis by stabilizing a certain conformation of Switch II (*Kupzig et al., 2006*; *Kupzig et al., 2009*).

RasGAPs such as p120GAP and neurofibromin also contain a proline at the position of Asn1005 in PlexinC1. GAP1m, the only GAP1 family member that is active toward Ras but not Rap, has a valine at this position (*Figure 11B*). Proline-to-valine mutants of the dual-specificity GAP1 family members (Rasal, CAPRI and GAP1$^{IP4BP}$) remain active toward Ras, but lose activity toward Rap (*Kupzig et al., 2009*). The superimposition of the p120GAP/Ras and PlexinC1/Rap structures suggests the basis for how this residue determines the substrate specificity of these GAPs (*Figure 11A*). Pro907 in p120GAP contributes to Ras binding by stacking against Tyr64 in Switch II of Ras. A valine residue at the position of Pro907 (Val515 in GAP1m in *Figure 11B*) appears to be readily accommodated in this Ras binding mode (*Figure 11A*). Assuming Rap adopts the same Gln63-in conformation when it binds the dual-specificity GAPs, a proline residue at this position in the GAPs is compatible with the interaction. However, the Gln63-in conformation of Rap places Phe64 much closer to the proline residue (*Figure 11A*). Replacing the proline with a bulkier valine residue likely cause steric clashes with Phe64 in Rap, leading to loss of the RapGAP activity.

## Unique interactions between plexin and Rap sharpen the specificity

In addition to Switch II, the PlexinC1/Rap interface involves several other residues in Rap that diverge from Ras/R-Ras/M-Ras. Residue 31 in Rap and Ras is a key residue for determining the binding specificity for downstream effectors of these two closely related small GTPases (*Nassar et al., 1996*). Our PlexinC1$_{cyto}$/Rap1B structure suggests that residue 31 is also a determinant for the specificity between the plexin GAP and Rap. Rap possesses a lysine at this position, which is replaced by a negatively charged residue (aspartate or glutamate) in Ras/R-Ras/M-Ras (*Figure 10A*). Lys31 and Asp33 in Rap form a charge–charge pair and are buried by the activation segment in PlexinC1 (*Figure 12B*). We made a Rap(K31E) mutant to render it more similar to Ras/R-Ras/M-Ras. This mutation is predicted to destabilize the PlexinC1/Rap interaction, since it closely places two buried negative charges. The GAP assay indeed showed that PlexinC1 failed to catalyze GTP hydrolysis for the K31E mutant (*Figure 12C*).

A potential salt-bridge between Asp95 in Rap1B and Lys666 in PlexinC1 may also contribute to their interaction and specificity (*Figure 12A*). Consistent with this notion, Rap2 has a proline residue at position 95 and is less responsive to the plexin GAP (*Wang et al., 2012*). The corresponding residues in Ras, R-Ras and M-Ras are glutamine, lysine and arginine respectively (*Figure 10A*). Mutating Rap Asp95 to lysine, as in R-Ras, substantially decreased the rate of PlexinC1-catalyzed GTP hydrolysis (*Figure 12C*). Likewise, PlexinC1(K666D) displayed lower GAP activity than the wild-type PlexinC1 (*Figure 12C*). The PlexinC1(K666D) and Rap(D95K) charge-swapped pair only slightly restored the GTP hydrolysis activity (*Figure 12C*), which may be due to disruption of the electrostatic complementarity

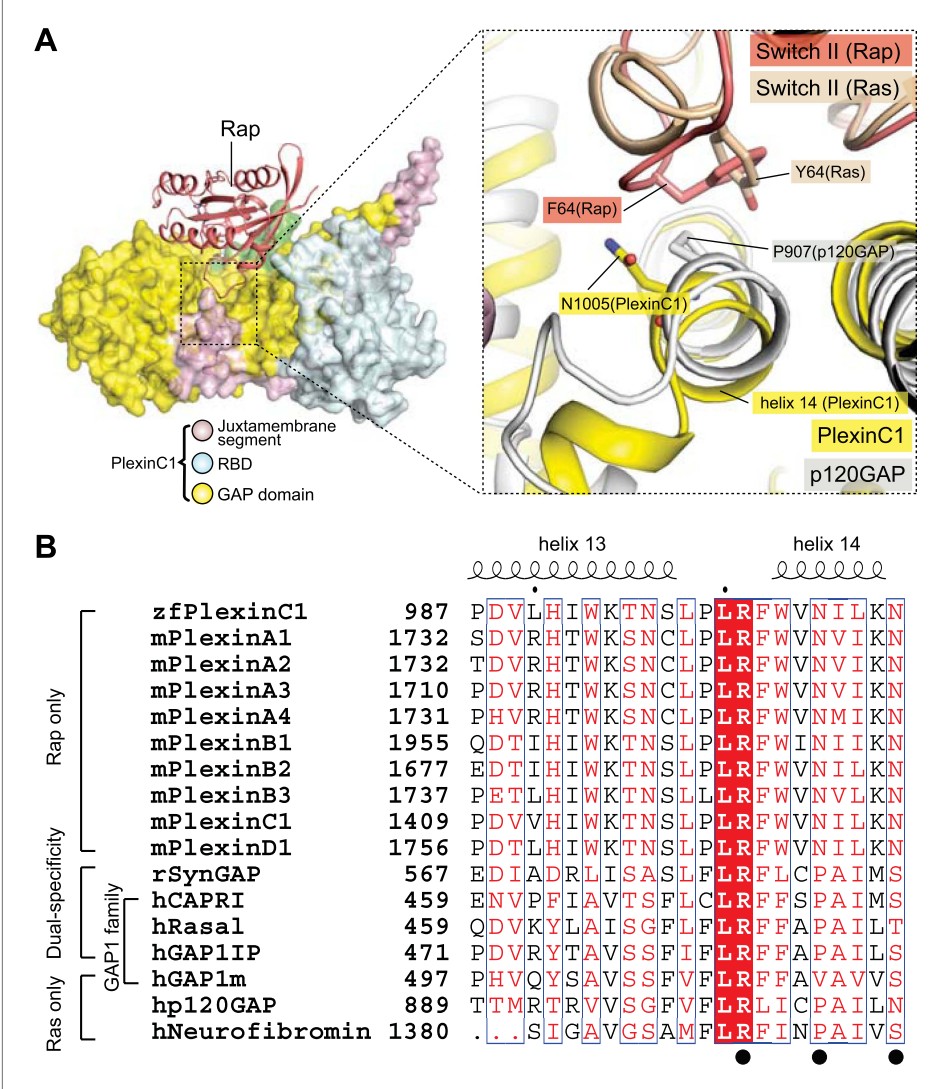

**Figure 11**. Comparison of the Switch II-interacting region between plexin, RasGAPs and dual-specific GAPs. (**A**) Packing interactions made by Phe64 in Rap1B with PlexinC1 and Tyr64 in Ras with p120GAP. The PlexinC1$_{ctyo}$/Rap1B and the p120GAP/Ras structures are superimposed by using Rap1B and Ras as references. (**B**) Sequence alignment of the major Switch II-interacting segment in plexins, RasGAPs and dual-specificity GAPs. The black circles highlight the three residues (Arg1001, Asn1005 and Asn1009) in zebrafish PlexinC1 that make critical interactions with Switch II of Rap. zf: zebrafish; m: mouse; h: human; r: rat.

at the plexin/Rap interface by the mutations. We also tested the importance of this interaction in the cell-based assay, which showed that the equivalent mutation of mouse PlexinA3 (R1360D) impaired the cell collapse activity (*Figure 12D*). The same residue in human PlexinA1 (Arg1384) has been found mutated to cysteine in cancer patients (*Seshagiri et al., 2012*). These analyses together with the unique plexin/Switch II interface support the notion that plexins have evolved to recognize residues in Rap that have diverged from other Ras family members, leading to loss of activity toward Ras/R-Ras/M-Ras.

## Concluding remarks

This study together with the previous structures of the plexin extracellular regions establishes a framework for understanding plexin regulation (*Figure 13*; *Video 1*) (*Janssen et al., 2010*; *Liu et al., 2010*; *Nogi et al., 2010*; *Janssen et al., 2012*). Semaphorin binding to the plexin extracellular region induces formation of the active dimer of the cytoplasmic region, which triggers its GAP activity to inactivate Rap through the non-canonical catalytic mechanism for signal transduction. Conformational

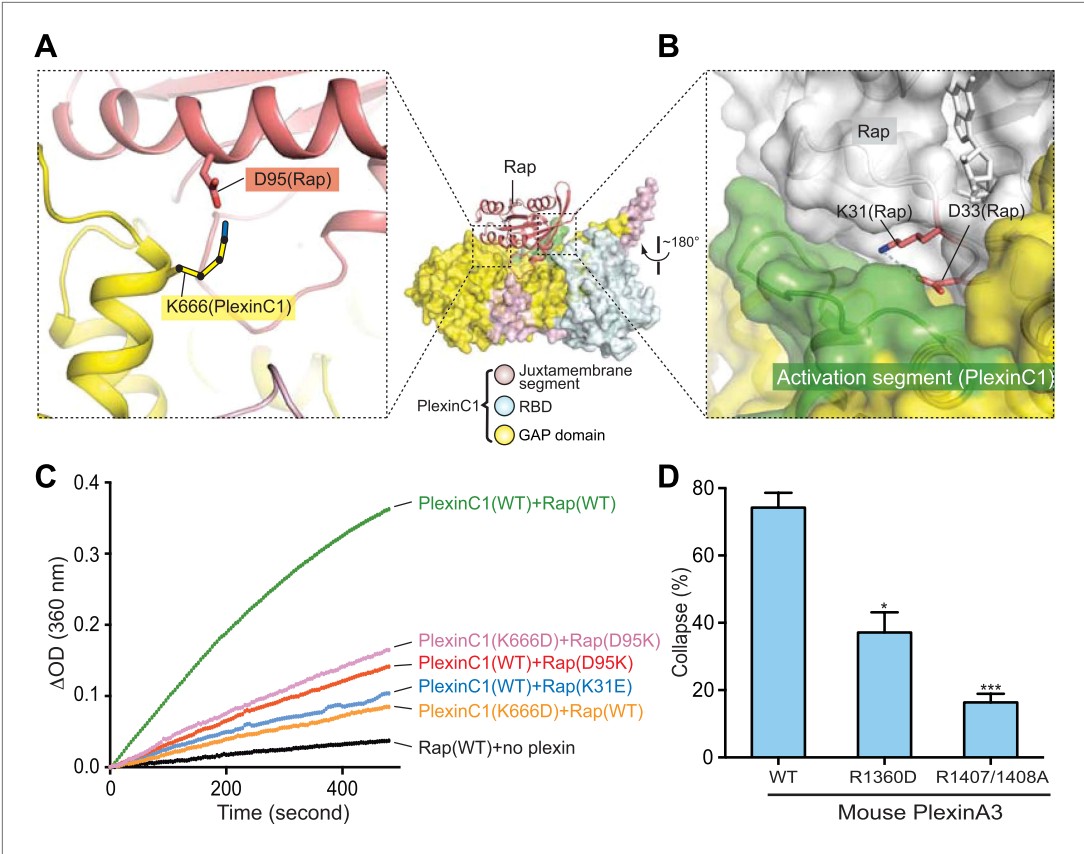

**Figure 12**. Additional specificity determinants in the PlexinC1/Rap1B complex. (**A**) Potential interaction between Lys666 in PlexinC1 and Asp95 in Rap. The side chain of Lys666 in PlexinC1 is not built in the final model due to weak electron density. It is modeled to show its potential interaction with Asp95 in Rap. (**B**) Burial of Lys31 in Rap1B at the PlexinC1/Rap1B interface. (**C**) GAP activity assays for the specificity determinants. Monomeric PlexinC1$_{cyto}$ was used in these assays. The plots are representatives of three replicates. (**D**) COS7 cell collapse assays for the R1360D mutant of mouse PlexinA3 (equivalent to K666D of zebrafish PlexinC1). The data analysis and presentation are the same as in **Figure 7D**.

changes similar to those undertaken by the plexin GAP domain upon the dimerization may serve as on/off switches for other related GAPs such as CAPRI, which is also activated by dimerization (***Dai et al., 2011***). In addition to activation of the GAP, the dimerization-induced structural rearrangements may underlie the activation state-selective binding of plexins by signal transducers such as FARP2 (FERM, RhoGEF and pleckstrin homology protein 2) and MICAL (molecule interacting with CasL) (***Toyofuku et al., 2005***; ***Schmidt et al., 2008***). The structures of the several extracellular membrane-proximal domains and the transmembrane helix of plexins have not been determined. Our data suggest that, upon semaphorin-induced dimerization, these domains are arranged precisely to ensure the proper juxtaposition of the juxtamembrane helix for inducing the active dimer of the cytoplasmic domain (***Figure 13***). Future work on these domains in the active dimeric state will fill in the missing links, leading to a complete structural model of semaphorin-activated plexin.

Clarifying the substrate specificity for the plexin GAP is essential for understanding plexin signaling. The results shown here and in our previous study (***Wang et al., 2012***) together demonstrate that while sharing the same domain fold with RasGAPs and dual-specificity GAPs, plexins are a unique group that are active to Rap, but not to Ras/R-Ras/M-Ras. Our analysis of the plexin/Rap complex structure reveals residues in both plexin and Rap that contribute to this specificity. P120GAP has been shown to bind GTP-bound Rap strongly but fail to catalyze its GTP hydrolysis, making Rap an effective inhibitor of the GAP activity of p120GAP to Ras/R-Ras/M-Ras (***Frech et al., 1990***; ***Hata et al., 1990***; ***Yatani et al., 1991***). The apparent GAP activity of plexins towards R-Ras and M-Ras reported previously may be caused indirectly by inactivation of Rap and alleviation of its inhibition on p120GAP. The induced Gln63-in

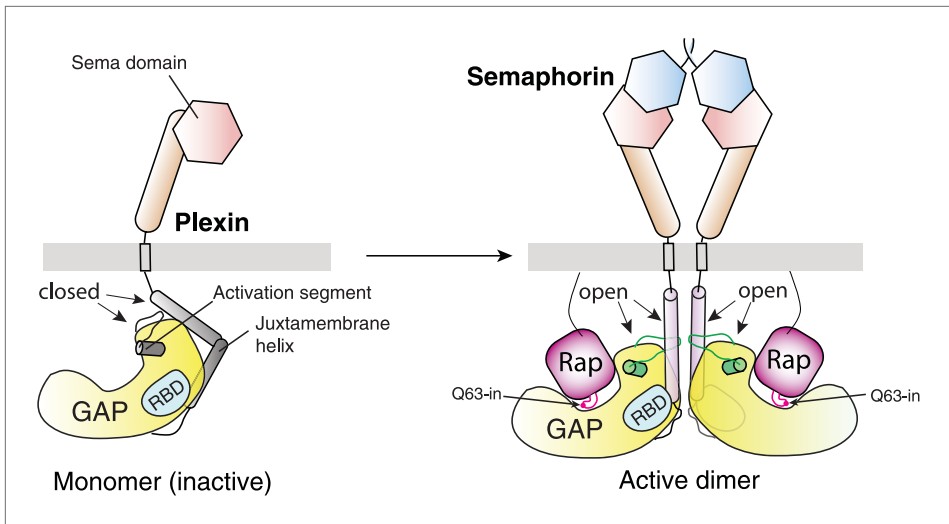

**Figure 13**. Schematic model for the activation of the plexin RapGAP by semaphorin-induced dimerization.

conformation of Rap seen in the PlexinC1/Rap complex structure likely represents the general mechanism by which plexins and the dual-specificity GAPs facilitate GTP hydrolysis for Rap. This conformation is stabilized by specific interactions made by several conserved residues in plexin. The dual-specificity GAPs achieve this through different mechanisms that likely involve the extra domains, the precise basis for which awaits structural studies of these GAPs in complex with Rap.

## Materials and methods

### Protein expression

The human Rap1B construct (residues 2–167) in a modified pET28 vector (Novagen, Darmstadt, Germany) that encodes a N-terminal His$_6$-tag and a recognition site for the human rhinovirus C3 protease has been described previously (**Wang et al., 2012**). The Rap1B constructs (2–166) containing a C-terminal flexible linker followed by a sortase recognition motif (one letter-code sequence: LPETGG) were generated by PCR and subcloned into the same vector. Seven versions of the linker were generated: 0-residue (containing the LPETGG motif only), 11-residue (sequence: GGSGGSGSGSS), 14-residue (sequence: SGGSGSGSSGGSGS), 16-residue (sequence: GGSGGSGSGSSGGSGS), 21-residue (sequence: G G S G G S G S G S S G G S G S GGGSG), 24-residue (sequence: SGGSGSGSSGG SGSGGGSGSGSSG) and 26-residue (sequence: GGSGGSGSGSSGGSGSGGGSGSGSSG). The vector encodes a glycine residue at the second position from the N-terminus, which becomes the N-terminal residue after removal of the methionine residue encoded by the start codon during protein expression. An N-terminal glycine on the Rap1B protein would hinder the sortase-mediated ligation with Plexin (see below) (**Popp et al., 2009**). To avoid this problem, the vector was mutated to replace the glycine residue with an aspartate using a

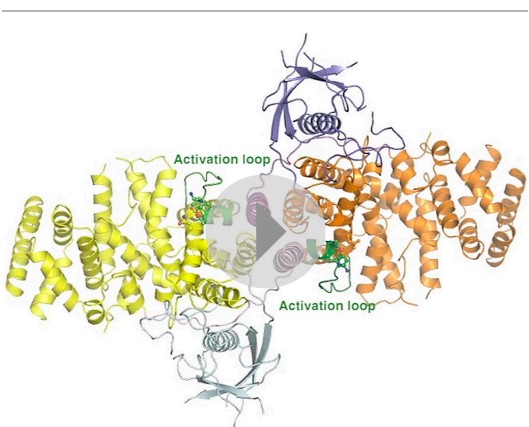

**Video 1**. Dimerization-induced activation of plexin$_{cyto}$ and binding of Rap to the GAP active site. The video is based on the crystal structures of inactive monomeric PlexinA3$_{cyto}$ (PDB ID: 3IG3), CC(a)PlexinC1$_{cyto}$ and the PlexinC1$_{cyto}$/Rap1B complex. It is rendered for illustrating the dimerization-induced structural rearrangements and the binding mode between plexin and Rap. The actual order of the events and conformational transition trajectories likely do not follow those in the video.

Quickchange reaction (Stratagene, La Jolla, CA). The Rap1B proteins were expressed in the bacteria strain BL21 (DE3) and purified as described previously (**Wang et al., 2012**).

The coding region for the zebrafish PlexinC1$_{cyto}$ (residues 552–1153) with a N-terminal di-glycine tag was synthesized (GenScript, Piscataway, NJ) based on the gene bank entry XM_685667.4. The region encoding residues 552–1147 was subcloned into another modified pET28 vector containing a N-terminal tandem His$_6$–SUMO tag (**Wang et al., 2012**). The GCN4 coiled-coil motif was fused to the N-terminus of the PlexinC1$_{cyto}$ (residues 553–1153) without the di-glycine motif by PCR. The fusion was subcloned into the same modified pET28 vector. Quikchange (Stratagene) was used to alter the residues at the junction between the coiled-coil and PlexinC1. The coiled-coil fusion constructs of mouse PlexinA1$_{cyto}$ were cloned by using similar procedures. The protein was expressed in the bacteria strain ArcticExpress (Stratagene) and purified as described previously (**He et al., 2009**; **Wang et al., 2012**). The His$_6$-SUMO-tag was removed by treatment with the SUMO-specific protease Ulp1. For the construct containing the di-glycine encoding sequence, the Ulp1 treatment yielded the PlexinC1 protein with a N-terminal GG-tag. All mutants of Rap and plexins were generated by Quickchange reactions (Stratagene), and expressed and purified as the respective wild-type proteins.

## Sortase-mediated ligation

Ligation of the N-terminal His$_6$/C-terminal LPETGG-tagged Rap1B and the N-terminal GG-tagged PlexinC1 was catalyzed by the transpeptidase activity of sortase from *Staphylococcus aureus* (plasmid provide by Dr Hidde Ploegh) (**Popp et al., 2009**). Sortase with a N-terminal His$_6$-tag was expressed and purified by using Ni-NTA chromatography. Sortase first cleaves the peptide bond between the threonine and first glycine within the LPETGG motif in Rap1B. In the second step, the GG-tagged PlexinC1 is added to the threonine to regenerate a native peptide bond between the two proteins. The reaction mix contained Rap1B, PlexinC1 and sortase at 450, 69 and 25 µM respectively. Reactions were performed at room temperature for 3 hr with simultaneous dialysis to remove the di-glycine by-product. The dialysis buffer contained 20 mM Tris pH 8, 150 mM NaCl, 10% glycerol, 2 mM MgCl$_2$, 2 mM DTT, 10 mM CaCl$_2$. The ligated PlexinC1/Rap1B complex was purified by Ni-NTA, ion exchange and gel filtration chromatographic steps. The N-terminal His$_6$-tag was removed by treatment with the human rhinovirus C3 protease.

## In vitro GAP assays

The GAP assay was performed by coupling release of inorganic phosphate during GTP hydrolysis to the purine nucleoside phosphorylase-catalyzed conversion of 2-amino-6-mercapto-7-methylpurine ribonucleoside to ribose-1-phosphate, which can be monitored photometrically at the wavelength of 360 nm (**Webb and Hunter, 1992**). For analyzing various structure-based mutations of CC(a)PlexinC1$_{cyto}$, the single turnover GAP assay was used (**Wang et al., 2012**). The concentration of plexin in the assays shown in **Figure 7C** was 0.25 µM. In the assays shown in **Figure 7A,B**, the concentration of plexin was 2 µM. The concentration of Rap1B(GTP) was 120 µM. In the assays for analyzing various mutants of the PlexinC1$_{cyto}$ monomer and Rap1B, the concentrations of PlexinC1 and Rap1B(GTP) were 5 µM and 60 µM respectively.

For determining the activation levels of the CC(x)Plexins$_{cyto}$ constructs, the initial reaction rate $V_0$ was measured at different Rap(GTP) concentrations ([S]) (**Table 2**). Fitting the data to the Michaelis–Menten equation ($V_0 = (V_{max}[S])/(K_M + [S])$) suggested that the Rap(GTP) concentrations used (25–150 µM) were far below $K_M$ (>1 mM). For plexin constructs exhibiting low GAP activity, $V_0$ was determined by linear fitting of the initial period of the reaction (5–8 min) when less then 10% of Rap(GTP) had been hydrolyzed. After subtraction of the baseline rate from reaction without plexin, the $k_{cat}/K_M$ value of each construct was estimated by fitting the data to the equation $V_0 = (k_{cat}/K_M) [E][S]$ (when $[S] << K_M$), where [E] is the total plexin concentration. For plexin constructs with high GAP activity, single turnover reaction curves measured at different Rap(GTP) concentrations were baseline-subtracted and simultaneously fitted to the single exponential equation: $A(t) = (A_{max} - A_{min}) (1 - exp(-kt)) + A_{min}$, where $k = (k_{cat}/K_M)[E]$. In the fitting, k was treated as a global parameter. The plexin and Rap concentrations and the analysis methods used are listed in **Table 2**.

Due to the high intrinsic activity of the ligated plexin$_{cyto}$/Rap complexes, all the bound GTP molecules were hydrolyzed to GDP during the purification process. To measure the GAP activity for these complexes, we used the multiple-turnover assay, in which $(NH_4)_2SO_4$ at 10 mM and EDTA at 1 mM are added to promote constant exchange of GTP/GDP for Rap in the complex, allowing continuous GTP hydrolysis provided sufficient GTP is present in the assay solution (**Webb and Hunter, 1992**).

**Table 2.** Protein concentrations and fitting methods used for determining $k_{cat}/K_M$ of plexins

| Plexin construct | Plexin concentration* (µM) | Rap-GTP concentrations (µM) | Data fitting method† |
|---|---|---|---|
| CC(a)PlexinA1$_{cyto}$ | 1.0 | 50.0, 75.0, 100.0, 150.0 | Single-exponential |
| CC(b)PlexinA1$_{cyto}$ | 5.0 | 50.0, 75.0, 100.0, 150.0 | Linear |
| CC(c)PlexinA1$_{cyto}$ | 5.0 | 50.0, 75.0, 100.0, 150.0 | Linear |
| CC(d)PlexinA1$_{cyto}$ | 1.0 | 50.0, 75.0, 100.0, 150.0 | Single-exponential |
| CC(e)PlexinA1$_{cyto}$ | 5.0 | 50.0, 75.0, 100.0, 150.0 | Single-exponential |
| CC(f)PlexinA1$_{cyto}$ | 5.0 | 50.0, 75.0, 100.0, 150.0 | Linear |
| CC(g)PlexinA1$_{cyto}$ | 1.0 | 50.0, 75.0, 100.0, 150.0 | Single-exponential |
| Monomer PlexinC1$_{cyto}$ | 2.0 | 25.0, 50.0, 75.0, 100.0, 150.0 | Linear |
| CC(a)PlexinC1$_{cyto}$ | 2.0 | 50.0, 75.0, 100.0, 150.0 | Single-exponential |
| CC(b)PlexinC1$_{cyto}$ | 2.0 | 25.0, 50.0, 75.0, 100.0, 150.0 | Linear |
| CC(d)PlexinC1$_{cyto}$ | 2.0 | 50.0, 75.0, 100.0, 150.0 | Single-exponential |
| CC(g)PlexinC1$_{cyto}$ | 2.0 | 25.0, 50.0, 75.0, 100.0, 150.0 | Linear |

*The plexin concentrations were chosen in order for the reaction rates to be within the dynamic range of the assay.
†Linear fitting: $k_{cat}/K_M$ determined by fitting data to $V_0 = (k_{cat}/K_M)$ [E][S]; Single-exponential fitting: $k_{cat}/K_M$ determined by fitting data to $A(t) = (A_{max} - A_{min})(1 - \exp(-kt)) + A_{min}$, in which $k = k_{cat}/K_M[E]$ and was fitted as a global parameter.

As the ligated complex with the 24-residue linker and the LPETGG motif crystallized and was used for structure determination, the same construct was chosen for extensive activity analyses at various concentrations. Unligated PlexinC1 and Rap1B mixed at the same concentrations were subjected to the same assay for comparison.

## Analytical ultracentrifugation

Sedimentation velocity analytical ultracentrifugation experiments were carried out using the ligated PlexinC1$_{cyto}$/Rap1B complex with the 24-residue linker and the LPETGG motif. Protein samples were prepared in Centrifugation Buffer (10 mM Tris pH 8, 50 mM NaCl, 2 mM TCEP, and 2 mM MgCl$_2$). Samples at 0.5, 4, and 20 µM were used for the experiments without AlF$_x$. Samples at 0.5, 4, 8 and 20 µM were used for the experiments in the presence of 2 mM AlF$_x$. All samples were equilibrated ~14 hr at 4°C, then ~400 µl of the samples were loaded into the 'sample' sides of dual-sectored charcoal-filled Epon centerpieces that were sandwiched between sapphire windows in a cell housing; the 'reference' sectors were filled with the same volume of Centrifugation Buffer. Filled cells were placed in an An50Ti rotor and equilibrated for 2.5 hr under vacuum in the centrifuge at 20°C prior to centrifugation. Experiments were conducted using a Beckman Optima XL-I analytical ultracentrifuge at 42,000 rpm at 20°C. Absorbance data at 280 nm were collected using the Beckman control software until all components had fully sedimented. Protein partial-specific volume, solvent viscosity, and density values were calculated using the program Sednterp (*Laue et al., 1992*). The data were analyzed using the $c(s)$ distribution in the program SEDFIT (*Schuck, 2000*). A regularization level of 0.68 was routinely employed. Time-invariant noise elements were removed from the data (*Schuck and Demeler, 1999*). Data-acquisition timestamp errors (*Zhao et al., 2013*) were examined with SEDFIT and were found to be ~0.1%; we deemed this small error acceptable and did not correct the timestamps. Plots were generated with the program GUSSI (http://biophysics.swmed.edu/MBR/software.html).

## Crystallization and structure determination

Mouse CC(d)PlexinA1, CC(g)PlexinA1, zebrafish CC(d)PlexinC1 and CC(a)PlexinC1 were subjected to crystallization trials. CC(a)PlexinC1$_{cyto}$ at 8 mg/ml crystallized initially at 20°C in 0.1M Bicine, pH 9.0, 20% PEG 6000 in sitting-drop 96-well plates. Larger crystals were grown by sitting-drop vapor diffusion at 20°C in 0.1M Bis-Tris propane, pH 9.1, 21% PEG 6000. Crystals were cryo-protected using the crystallization solution supplemented with 25% glycerol and flash cooled in liquid nitrogen. Diffraction data were collected at 100 K on beamline 19ID at the Advanced Photon Source (Argonne National

Laboratory). Data were indexed, integrated and scaled by using HKL2000 (*Otwinowski and Minor, 1997*). A 3.3 Å dataset in the $P2_12_12_1$ space group was collected. The 'autocorrections' option in HKL2000 was selected to truncate and scale the anisotropic data, which was then converted to the mtz format by using the Ctruncate program in CCP4 (*Padilla and Yeates, 2003*; *Winn et al., 2011*). The structure of the GAP domain of mouse PlexinA3 (PDB ID: 3IG3) was used as the molecular replacement search model using the Phaser module in the Phenix package (*Adams et al., 2002*; *Mccoy et al., 2007*).

Ligated complexes of human Rap1B and several plexins$_{cyto}$ from mouse and zebrafish each with one of the 7 versions of the linker mentioned above were all subjected to crystallization trails. The ligated complex of zebrafish PlexinC1$_{cyto}$ and human Rap1B with the 24-residue linker and the LPETGG-tag at 4 mg/ml crystallized initially at 20°C in 0.1 M HEPES pH 7.5, 10% 2-propanol, 20% PEG 4K in sitting-drop 96-well plates. Larger crystals were grown by hanging-drop vapor diffusion at 20°C in 0.1 M HEPES pH 7.3, 5% 2-propanol, 25% PEG 3350, 3.6% polypropylene glycol P400. Cryo-protection of the crystals was achieved using with the crystallization solution supplemented with 25% glycerol. Cryo-protected crystals were snap cooled in liquid nitrogen. The data collection and processing were performed in a similar manner as described for the CC(a)PlexinC1$_{cyto}$ crystal, expect that the 'autocorrections' option was not used. The diffract pattern extended to 3.3 Å and was consistent with the symmetry of the P1 space group. One protomer from the CC(a)PlexinC1$_{cyto}$ structure was used as the molecular replacement search model for plexin. The structure of Rap1B from the Rap1B/RapGAP complex (PDB ID: 3BRW) was used as the search model for Rap1B.

Iterative model building and refinement were performed using the Phenix and Coot programs respectively (*Adams et al., 2002*; *Emsley and Cowtan, 2004*). In the PlexinC1$_{cyto}$/Rap1B structure, the linker between the C-terminus of Rap1B and the N-terminus of PlexinC1$_{cyto}$ is not included in the final model due to lack of discernable electron density. Assuming the complexes in the crystal are formed by the covalently linked pairs of Rap1B and PlexinC1$_{cyto}$, the linker and the disordered flanking residues from the two proteins (a total of ~32 residues) are sufficient for spanning the ~45 Å distance between the two ends without imposing restraints on the plexin/Rap binding mode (*Figure 8A*, middle panel). The structural superimpositions shown in the *Figures 5 and 6* were based on helices 13, 14 and 15 in the plexin GAP domain, because they are at the center of the GAP active site and adopt highly similar conformations in all the plexin structures. Comprehensive model validation was performed by using MolProbity (*Chen et al., 2010*). Detailed statistics of data collection and refinement are listed in *Table 1*. Structure figures were prepared in PyMOL (the PyMOL Molecular Graphics System, Schrodinger). Sequences were aligned by using T-Coffee (*Notredame et al., 2000*) and rendered with ESPript (*Gouet et al., 1999*). Molecular surface area was calculated using the get_area function in PyMOL. Morph frames in *Video 1* were generated by using the Yale morph server (*Krebs and Gerstein, 2000*) and rendered in PyMOL.

## COS7 cell collapse assay

Mutants of mouse PlexinA3 were designed based on a sequence alignment of zebrafish PlexinC1 with all mouse plexins (Plexin A1, A2, A3, A4, B1, B2, B3, C1 and D1). COS7 cell collapse assays using full-length mouse PlexinA3 were performed as described previously (*He et al., 2009*). Briefly, $1 \times 10^5$ COS7 cells were plated in each well of a 6-well plate one day prior to transfection. FuGENE 6 (Promega, Madison, WI) was used to transfect each well with PlexinA3 (1 μg plasmid) and the co-receptor Neuropilin2 (0.5 μg plasmid) following the manufacturer's instructions. 2 days post transfection, 5 nM alkaline phosphatase-tagged Sema3F was added to each well and incubated for 25 min at 37°C. The cells were washed, fixed and heat-treated at 65°C for 1 hr to inactivate endogenous phosphatases. Cells were stained with the BCIP/NBT alkaline phosphatase substrate (Sigma, St. Louis, MO), and counted using a randomized and blind method.

## Acknowledgements

We thank Andrew Schober, Josee Santiago and members in the Zhang laboratory for discussions and technical assistance. We also thank Dr Hidde Ploegh at the Whitehead Institute for providing the sortase construct, Dominika Borek, Zhe Chen, the staff at the structural biology laboratory at UTSW and at beamline 19ID of Advanced Photon Source for assistance with X-ray data collection and processing. XZ is a Virginia Murchison Linthicum Scholar in Medical Research at UTSW. Results shown in this report are derived from work performed at Argonne National Laboratory, Structural Biology Center at the

Advanced Photon Source. Argonne is operated by U. Chicago Argonne, LLC, for the US Department of Energy, Office of Biological and Environmental Research under contract DE-AC02-06CH11357.

## Additional information

### Funding

| Funder | Grant reference number | Author |
|---|---|---|
| National Institutes of Health | GM088197 | Yuxiao Wang, Heath G Pascoe, Huawei He, Xuewu Zhang |
| The Welch foundation | I-1702 | Yuxiao Wang, Huawei He, Xuewu Zhang |
| National Institutes of Health | GM008203 | Heath G Pascoe |

The funders had no role in study design, data collection and interpretation, or the decision to submit the work for publication.

### Author contributions

YW, HGP, XZ, Conception and design, Acquisition of data, Analysis and interpretation of data, Drafting or revising the article; CAB, Acquisition of data, Analysis and interpretation of data, Drafting or revising the article; HH, Conception and design, Acquisition of data, Drafting or revising the article

## Additional files

### Major dataset

The following datasets were generated:

| Author(s) | Year | Dataset title | Dataset ID and/or URL | Database, license, and accessibility information |
|---|---|---|---|---|
| Pascoe HG, Zhang X | 2013 | Crystal structure of the PlexinC1/Rap1B complex | http://www.rcsb.org/pdb/explore/explore.do?structureId=4M8N | Publicly available at the Protein Data Bank (http://www.rcsb.org/pdb/). |
| Wang Y, Zhang X | 2013 | Crystal structure of the active dimer of zebrafish PlexinC1 cytoplasmic region | http://www.rcsb.org/pdb/explore/explore.do?structureId=4M8M | Publicly available at the Protein Data Bank (http://www.rcsb.org/pdb/). |

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
