## [Decision Letter]

Thank you for choosing to send your two studies entitled “Structural basis of dimerization-induced activation of the plexin cytoplasmic region” and “Crystal structure of a plexin/Rap complex reveals the non-canonical catalytic mechanism for Rap inactivation” for consideration at *eLife*. Your articles have been favorably evaluated by a Senior editor and 2 reviewers, one of whom, Axel Brunger, is a member of our Board of Reviewing Editors.

The Reviewing editor and the other reviewers discussed their comments before we reached this decision, and the Reviewing editor has assembled the following comments to help you prepare a revised submission.

The crystal structures (PlexinC1/Rap/ADP AlFx complex and coiled-coil induced homo-dimer of PlexinC1) presented in two manuscripts reveal a new dimerization-dependent mechanism of the GAP activity of PlexinC1.

The structure with Rap1B bound to the plexinC1 cytoplasmic domain in complex with ADP AlFx revealed that plexin induces a conformational change in the Switch II region of Rap, bringing Glu63 into the active site to catalyze GTP hydrolysis. There is some concern about the medium resolution of this structure, but the interactions found were validated by mutagenesis studies of GAP activity.

Rap1B binding induced a symmetric dimer of plexinC1. The plexinC1 cytoplasmic domain, when dimerized by fusion with an N-terminal coiled-coil domain to mimic ligand binding, formed a nearly identical symmetrical back-to-back dimer.

Overall, the formation of the plexinC1 dimer led to a major movement of the juxtamembrane segment compared to the known monomeric plexinC1 form, along with an outward shift of the loop-helix segment in the GAP domain. It is this induced conformation that allows Rap binding and GTP hydrolysis. Mutational analysis of key residues in the dimer interface supports the authors’ conclusion that this mode of dimerization leads to an increase in Rap GAP activity. Taken together, these new structures, along with the associated biochemical studies, reveal for the first time how plexin cytoplasmic domains signal in response to semaphorin binding, through dimerization-mediated conformational changes in the cytoplasmic domain promoting Rap GTP binding and induced GTP hydrolysis.

Required revisions:

1) Both articles: there is consensus among all reviewers of these manuscripts that the two need to be combined into one manuscript, as the structure of the coiled-coil induced dimer of PlexinC1 appears to be very similar to that of PlexinC1 in the heterodimer with Rap. As it stands, there is too much redundancy between the two articles, and they both depend on each other to support the key conclusions. The merging of the two manuscripts into one will make the story much easier to read and understand.

2) Both articles: both structures used insightful methods to accomplish crystallization – fusion to a coiled coil protein and ligation between two proteins, respectively. The authors have done some controls, such as a GAP activity assays and confirming that the observed interactions are supported by mutagenesis studies. Also, the similarity of the PlexinC1 dimer in both structures is reassuring. However, it would be interesting to examine the structures themselves to see if the coiled-coil domain may have affected or “strained” the dimer structure somehow. Also, the authors might comment on any general lessons they learned in using the fusion and ligation approaches (also considering that the linker spanned ∼45 Å), respectively, that might be useful for other systems as well.

3) Complex article: is it definitively AlF3 rather than another coordination species?

4) Both articles: how is GTP hydrolysis in the GAP assay exactly measured?

5) *eLife* discourages the use of supplemental figures and tables unless the supplemental material consists of raw data, or repetitions of information shown in main figures. Thus, all supplemental figures need to be fully merged with figures in the main body of the text. However, all reviewers noted a lot of redundancy among the various figure panels and too much unnecessary detail, so in combining the figures, the authors should work on the presentation of their structures in as few figures as possible. Moreover, sequence alignments could be omitted altogether as long as it is clearly stated how they were obtained, and not essential to the paper.

6) Dimer article: all supplemental figures should be moved to the main body of the paper as separate figures for better readability.

7) Both manuscripts: Table 1. Please provide the complete validation reports provided by the PDB.

8) Complex article: please comment on the relatively high R-free value of the complex structure relative to the resolution of that structure.

9) Dimer article: the section on the lack of change in the conformation of the RBD and the distal part of the GAP domain could be contracted into a single sentence stating that there are no differences of interest.

10) Dimer article: the labeling and legend of Figure 1 are more complicated than necessary. Identical pairs of coiled-coil diagrams do not need to be shown 11 times. They can be shown once and the corresponding entry on the graph labeled “a”, “b”, etc. Numbers are missing on the x-axis.

11) Dimer article: Figure 2 is overall quite clear but could be improved by labeling the juxtamembrane helix and 11/11' in (B), and the activation segment in both (B) and (C).

12) Dimer article: Figure 4, the steric collisions shown in panel B are not that dramatic as shown. Perhaps the D38-N1774 collision would be clearer if those were shown in space-filling form. Overall, as depicted in this figure, the autoinhibition only barely obstructs the Rap binding site. This might be worth some additional discussion.

13) Complex article: with respect to the overall impact of the study, various permutations of Gln fingers, dual Arg and Gln fingers, and Asn thumbs have appeared in the small GTPase GAP literature over the past 11 years or so, and this work should be discussed and referenced.

---

## [Author Response]

*1) Both articles: there is consensus among all reviewers of these manuscripts that the two need to be combined into one manuscript, as the structure of the coiled-coil induced dimer of PlexinC1 appears to be very similar to that of PlexinC1 in the heterodimer with Rap. As it stands, there is too much redundancy between the two articles, and they both depend on each other to support the key conclusions. The merging of the two manuscripts into one will make the story much easier to read and understand*.

We have merged the two papers as suggested.

*2) Both articles: both structures used insightful methods to accomplish crystallization – fusion to a coiled coil protein and ligation between two proteins, respectively. The authors have done some controls, such as a GAP activity assays and confirming that the observed interactions are supported by mutagenesis studies. Also, the similarity of the PlexinC1 dimer in both structures is reassuring. However, it would be interesting to examine the structures themselves to see if the coiled-coil domain may have affected or “strained” the dimer structure somehow). Also, the authors might comment on any general lessons they learned in using the fusion and ligation approaches (also considering that the linker spanned ∼45 Å), respectively, that might be useful for other systems as well*.

We thank the reviewers for pointing this out. Indeed there is a small difference in the N-terminal portion of the juxtamembrane helix between the two structures, which is likely caused by a small “geometric incompatibility” between the coiled-coil and the plexin dimer. We have included a detailed discussion on this point in the paper. Regarding the lessons in using the fusion and ligation approaches, it is hard to come up with some general rules other than the notion that these approaches are useful for crystallizing protein dimers or complexes with weak affinities. We tested many linkers on a trial-and-error basis, most of which were long enough to bridge the distance between plexin and Rap. Yet, only the 24-residue linker construct led to crystallization.

*3) Complex article: is it definitively AlF3 rather than another coordination species*?

While the relative low resolution prevents definitive identification of the form of AlFx, the density is most consistent with the trigonal AlF3. In addition, the highly similar geometry at the active site between our structure and the p120GAP/Ras complex also suggest it is AlF3. We have included this discussion in the paper.

*4) Both articles: How is GTP hydrolysis in the GAP assay exactly measured*?

The GAP assay was performed by coupling release of inorganic phosphate during GTP hydrolysis to purine nucleoside phosphorylase-catalyzed conversion of 2-amino-6-mercapto-7-methylpurine ribonucleoside to ribose-1-phosphate, which can be monitored photometrically at the wavelength of 360 nm (68). We have included this description in the Materials and methods section.

*5)* eLife *discourages the use of supplemental figures and tables unless the supplemental material consists of raw data, or repetitions of information shown in main figures. Thus, all supplemental figures need to be fully merged with figures in the main body of the text. However, all reviewers noted a lot of redundancy among the various figure panels and too much unnecessary detail, so in combining the figures, the authors should work on the presentation of their structures in as few figures as possible. Moreover, sequence alignments could be omitted altogether as long as it is clearly stated how they were obtained, and not essential to the paper*.

Along with merging of the two papers, we have completely reorganized the figures and eliminated supplemental figures as suggested. We have removed the sequence alignment figure (the original Figure 5 of the dimer paper). We did keep the sequence alignments showing residue differences in the GAPs and the small GTPases that are key to determination of the specificity.

*6) Dimer article: All supplemental figures should be moved to the main body of the paper as separate figures for better readability*.

Done as suggested; see above.

*7) Both manuscripts:*
Table 1*. Please provide the complete validation reports provided by the PDB*.

The structures have been deposited in the RCSB database (PDB IDs: 4M8M and 4M8N) and are currently under processing. They will be released upon publication of the paper. The validation reports have been provided.

*8) Complex article: please comment on the relatively high R-free value of the complex structure relative to the resolution of that structure*.

Despite extensive optimization, the best complex crystal we obtained still displayed streaky diffraction pattern and high mosaicity. The P1 space group and the weak diffraction limited the completeness and redundancy of the data. All these contributed to the relatively poor data quality, which likely explains the relatively high R-free value. Regardless, the main conclusion from this structure is not affected because the binding interface between plexin and Rap is among the most well-ordered parts in the structure and show clear electron density (see the simulated annealing omit map in Figure 8, right panel).

*9) Dimer article: the section on the lack of change in the conformation of the RBD and the distal part of the GAP domain could be contracted into a single sentence stating that there are no differences of interest*.

This has been done as suggested.

*10) Dimer article: the labeling and legend of*
Figure 1
*are more complicated than necessary. Identical pairs of coiled-coil diagrams do not need to be shown 11 times. They can be shown once and the corresponding entry on the graph labeled “a”, “b”, etc. Numbers are missing on the x-axis*.

We have changed the figure and labels as suggested.

*11) Dimer article:*
Figure 2
*is overall quite clear but could be improved by labeling the juxtamembrane helix and 11/11' in (B), and the activation segment in both (B) and (C)*.

This has been done.

*12) Dimer article:*
Figure 4*, the steric collisions shown in panel B are not that dramatic as shown. Perhaps the D38-N1774 collision would be clearer if those were shown in space-filling form. Overall, as depicted in this figure, the autoinhibition only barely obstructs the Rap binding site. This might be worth some additional discussion*.

We agree that the clash caused by the autoinhibited state appears not very severe. As discussed in the paper, the disordered portion of the activation segment may also contribute to autoinhibition by sampling conformations that obstruct Rap binding, whereas in the dimer structure this segment is stabilized in the open conformation. These effects together may account for the activity difference between the active dimer and the monomer (which displays basal activity and is not completely inactive). In addition, interactions between small GTPases and their regulators seem to be rather sensitive to subtle changes at the interface. For example, a single Leu/Ile in Dbl-family RhoGEFs serves as the major specificity determinant for distinguishing Rac1 and Cdc42 (Snyder, Nature Structural & Molecular Biology, 2002, 9, 468). We have included this discussion in the paper.

*13) Complex article: With respect to the overall impact of the study, various permutations of Gln fingers, dual Arg and Gln fingers, and Asn thumbs have appeared in the small GTPase GAP literature over the past 11 years or so, and this work should be discussed and referenced*.

In addition to the original description in the Introduction, we have included one more sentence and one new reference to summarize these previous findings and put our work into the context. We refrained from more extensive discussion on this since the merged paper has become rather long.